# TIGER: Time-frequency Interleaved Gain Extraction and Reconstruction for Efficient Speech Separation

**Mohan Xu**[1,*]**, Kai Li**[1,2,*]**, Guo Chen**[1,2] **& Xiaolin Hu**[1,2,3,†]
1. Department of Computer Science and Technology, Institute for AI,
BNRist, Tsinghua University, Beijing 100084, China
2. Tsinghua Laboratory of Brain and Intelligence (THBI),
IDG/McGovern Institute for Brain Research, Tsinghua University, Beijing 100084, China
3. Chinese Institute for Brain Research (CIBR), Beijing 100010, China
`xu-mh19@tsinghua.org.cn`
`{li-k24, cg22}@mails.tsinghua.edu.cn`
`xlhu@tsinghua.edu.cn`

## Abstract

In recent years, much speech separation research has focused primarily on improving model performance. However, for low-latency speech processing systems, high efficiency is equally important. Therefore, we propose a speech separation model with significantly reduced parameters and computational costs: *Time-frequency Interleaved Gain Extraction and Reconstruction network (TIGER)*. TIGER leverages prior knowledge to divide frequency bands and compresses frequency information. We employ a multi-scale selective attention module to extract contextual features, while introducing a full-frequency-frame attention module to capture both temporal and frequency contextual information. Additionally, to more realistically evaluate the performance of speech separation models in complex acoustic environments, we introduce a dataset called *EchoSet*. This dataset includes noise and more realistic reverberation (e.g., considering object occlusions and material properties), with speech from two speakers overlapping at random proportions. Experimental results showed that models trained on EchoSet had better generalization ability than those trained on other datasets to the data collected in the physical world, which validated the practical value of the EchoSet. On EchoSet and real-world data, TIGER significantly reduces the number of parameters by **94.3**% and the MACs by **95.3**% while achieving performance surpassing state-of-the-art (SOTA) model TF-GridNet.

## 1 Introduction

Humans possess the ability to focus on a specific speech signal in noisy environments, which is a phenomenon known as the "cocktail party effect" (Cherry, 1953). In speech processing, the corresponding challenge is accurately separating different sound sources from mixed audio signals, a task referred to as speech separation. Speech separation is typically used as a preprocessing step for speech recognition, as it helps enhance recognition accuracy (Haykin & Chen, 2005). Consequently, it is crucial to ensure that speech separation not only produces clear and distinct outputs on real-world audio but also meets the demand of high computational efficiency (Divenyi, 2004).

In recent years, the application of deep learning methods to the speech separation task has received widespread attention (Wang et al., 2023; Li et al., 2023; 2022; Li & Luo, 2023; Subakan et al., 2021). Although many high-performing speech separation methods have been proposed, two key issues remain insufficiently addressed.

First, when designing a separation model, we should fully take into account the actual application scenarios of the speech processing system, which require low latency and low computational

---

*Mohan Xu and Kai Li contribute equally to the article. [†] Corresponding author.

complexity. However, many approaches have primarily focused on improving speech separation performance. For example, TF-GridNet (Wang et al., 2023) utilizes bidirectional LSTMs and self-attention mechanisms in an alternating manner, achieving good results on benchmark datasets but have large model size. To make the separation model more applicable in computationally constrained real-world scenarios, TDANet (Li et al., 2023) introduces an efficient lightweight architecture using top-down attention, achieving competitive performance with lower computational costs than SepFormer (Subakan et al., 2021). However, as a time-domain method, TDANet struggles to leverage frequency-domain information. On the other hand, time-frequency domain approaches like TF-GridNet (Wang et al., 2023) model both time and frequency dimensions but require higher computational resources. BSRNN (Luo & Yu, 2023), which is the SOTA model for music separation, reduces the computational burden by focusing on important frequency bands. The band-split strategy is enlightening but under-explored in speech separation. How to balance computational efficiency and separation quality in speech separation is still a big challenge.

Second, the commonly used speech separation datasets exhibit a significant gap from real-world scenarios. Many methods relied on the WSJ0-2mix dataset (Hershey et al., 2016) for evaluation, which only contains fully-overlapping audio without noise or reverberation. Models trained on this kind of dataset are subject to weak generalization and robustness in real-world environments (Kadıoğlu et al., 2020; Cosentino et al., 2020). Although the WHAMR! dataset (Maciejewski et al., 2020) adds noise and reverberation to WSJ0-2mix, the generated reverberation fails to fully take into account factors such as object occlusion and material properties, and the diversity of acoustic scenarios remains limited. To more accurately train and evaluate speech separation models for practical use, a dataset that more closely resembles real-world environments is necessary. Specifically, this dataset should include different noise types, cover a wide range of realistic acoustic environments, and have randomly distributed speech overlap ratios.

To address the two issues mentioned above, our main contributions are as follows:

1. We propose a novel lightweight separation model named TIGER. TIGER adopts a band-split strategy to reduce computational costs by leveraging prior knowledge in the frequency domain. Furthermore, TIGER introduces the frequency-frame interleaved (FFI) block, composed of two key submodules: multi-scale selective attention (MSA) and full-frequency-frame attention ($F^3A$). These submodules enable efficient integration of temporal and frequency features.

2. We propose a speech separation dataset called EchoSet. It is a high-fidelity dataset bridging the gap between model training and real-world applications.

Experiments show that models trained on EchoSet generalized better on real-world data than those trained on benchmark dataset LRS2-2Mix (Li et al., 2023) and Libri2Mix (Cosentino et al., 2020), validating that audio in EchoSet is closer to the physical world. We then comprehensively evaluated TIGER on Libri2Mix, LRS2-2Mix and EchoSet. As the dataset becomes more complex, TIGER's superiority in performance becomes more significant. On EchoSet, which is the most complicated among the three datasets, TIGER improved the performance by about 5% compared with TF-GridNet, while reducing the parameters and MACs by 94.3% and 95.3% respectively. When tested on real-world data, TIGER also achieved the best separation performance. The remarkable result shows that TIGER provides a new solution for the design of lightweight speech separation models for practical use in the time-frequency domain.

## 2 RELATED WORK

**Speech separation**. Speech separation methods can be divided into time domain and time-frequency domain. Time domain methods directly process the original audio signal. Conv-TasNet (Luo & Mesgarani, 2019) extracts features by temporal convolutional network (Lea et al., 2016). To improve the performance on long sequence data, DPRNN (Luo et al., 2020) divides the temporal sequence into small blocks and alternately performs intra-block and inter-block modeling, which becomes a common paradigm for many following works (Wang et al., 2023; Subakan et al., 2021). Time-frequency domain methods apply a Short-Time Fourier Transform (STFT) to transform the waveform into a joint representation of time and frequency. TF-GridNet (Wang et al., 2023) enhances the temporal context information by a full-band self-attention module. Although TF-GridNet achieved SOTA performance, it entails huge computational costs.

**Lightweight models**. Some models (Wang et al., 2023; Yang et al., 2022; Subakan et al., 2021) with high computational complexity are difficult to be applied to real-time speech processing on edge devices. To reduce computational costs, TDANet (Li et al., 2023) draws on the attention mechanism of human brains and designs a lightweight structure. In music separation, BSRNN (Luo & Yu, 2023) uses prior knowledge to split band, performing band merging on less important bands to compress the feature while retaining key band information.

**Datasets for speech separation**. WSJ0-2mix (Hershey et al., 2016) is an early and commonly used fully-overlapping clean speech separation dataset. WHAM! (Wichern et al., 2019) added environmental noises to WSJ0-2mix, and furthermore WHAMR! (Maciejewski et al., 2020) added simple reverberation. Libri2Mix (Cosentino et al., 2020) was proposed based on the observation (Kadıoğlu et al., 2020) that the test performance of Conv-TasNet trained on WSJ0-2mix dropped sharply on other separation datasets. The utterances in Libri2Mix were mixed with sparse overlap, and noises were added to the mixed audio, but reverberation was not considered in Libri2Mix. LRS2-2Mix (Li et al., 2023) was mixed by video clips acquired through BBC. The audio was recorded in real acoustic scenarios, thus containing much noise and reverberation. However, due to the different recording environments of the clips, such as the shapes and materials of the room and objects, the reverberation obtained when the clips were directly mixed is still unrealistic.

## 3 TIGER

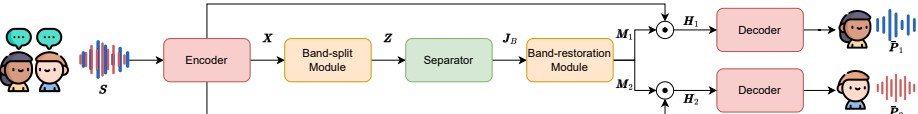

Figure 1: The overall pipeline of TIGER. We focus on scenarios with only two speakers.

### 3.1 OVERALL PIPELINE

Let $L$ be the sequence length of an audio. Given a monaural mixture audio $\boldsymbol{S} \in \mathbb{R}^{1 \times L}$ containing utterances of $C$ speakers and noise $\boldsymbol{n} \in \mathbb{R}^{1 \times L}$:

$$\boldsymbol{S} = \sum_{i}^{C} \boldsymbol{P}_i + \boldsymbol{n}, \tag{1}$$

the speech separation task is to recover the clean speech of each speaker $\boldsymbol{P}_i \in \mathbb{R}^{1 \times L}$.

The TIGER system (Figure 1) can be divided into five main components: the encoder, the band-split module, the separator, the band-restoration module, and the decoder. Specifically, we first use STFT as the encoder to convert the mixed audio signal $\boldsymbol{S} \in \mathbb{R}^{1 \times L}$ into its time-frequency representation $\boldsymbol{X} \in \mathbb{C}^{F \times T}$, where $F$ and $T$ represent the number of frequency bins and time frames, respectively. Next, we apply a frequency band-split strategy, dividing the whole band into $K$ sub-bands of varying widths based on their importance. Each sub-band is transformed into a uniform channel size $N$ using 1D convolutions, and these are then stacked along the frequency dimension to produce the feature representation $\boldsymbol{Z} \in \mathbb{R}^{N \times K \times T}$. Thirdly, $\boldsymbol{Z}$ serves as the input to the separator, which uses FFI blocks with shared parameters to model the acoustic characteristics of each speaker. Subsequently, the band-restoration module restores the sub-bands to the full frequency range using separator output $\boldsymbol{J}_B \in \mathbb{R}^{N \times K \times T}$ ($B$ denotes number of blocks in separator), and the mask for each speaker $\boldsymbol{M}_i \in \mathbb{C}^{F \times T}$ is applied element-wise product to $\boldsymbol{X}$, producing the separated representation for each speaker $\boldsymbol{H}_i \in \mathbb{C}^{F \times T}$. Finally, the inverse STFT is used to generate the clean speech signal $\bar{\boldsymbol{P}}_i \in \mathbb{R}^{1 \times L}$ for each speaker.

### 3.2 BAND-SPLIT MODULE

Given a time-frequency representation $\boldsymbol{X}$, we first apply a frequency band-split strategy to divide the frequency dimension into $K$ frequency sub-bands $\{\boldsymbol{B}_k \in \mathbb{C}^{G_k \times T} | k = [1, K]\}$ :

$$F = \sum_{k=1}^{K} G_k. \tag{2}$$

The widths of the sub-bands $G_k$ are not necessarily the same. For each frequency sub-band $\boldsymbol{B}_k$, we merge its real $\text{Re}(\cdot)$ and imaginary $\text{Im}(\cdot)$ parts into the frequency dimension to generate $\dot{\boldsymbol{B}}_k \in \mathbb{R}^{2G_k \times T}$. We denote the concatenation operation as $||$, then:

$$\dot{\boldsymbol{B}}_k = \text{Re}(\boldsymbol{B}_k)||\text{Im}(\boldsymbol{B}_k). \tag{3}$$

Next, we transform the frequency dimension $2G_k$ of $\dot{\boldsymbol{B}}_k$ to the feature dimension $N$ using a group normalization layer followed by a 1D convolution, which utilizes a kernel size of 1 and does not share parameters across different $\dot{\boldsymbol{B}}_k$. In this way, we obtain feature $\boldsymbol{Z}_k \in \mathbb{R}^{N \times T}$ of the same shape for each sub-band. We then stack the features $\boldsymbol{Z}_k$ from the $K$ frequency sub-bands along the frequency dimension to yield the input feature $\boldsymbol{Z} \in \mathbb{R}^{N \times K \times T}$ for the separator.

## 3.3 SEPARATOR

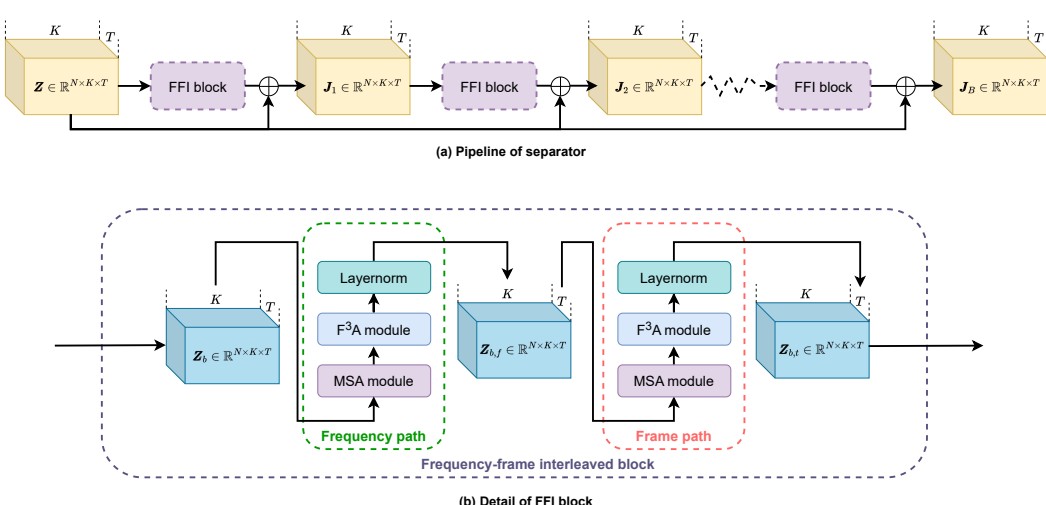

Figure 2: The separator of TIGER, consists of several FFI blocks which share parameters. Residual connections are used to retain original features and reduce learning difficulty.

In the separator, the input feature $\boldsymbol{Z}$ passes sequentially through $B$ frequency-frame interleaved (FFI) blocks with shared parameters, as shown in Figure 2. In each FFI block, the frequency path is first used to extract contextual information between different sub-bands, producing $\boldsymbol{Z}_{b,f} \in \mathbb{R}^{N \times K \times T}$. Next, we feed $\boldsymbol{Z}_{b,f}$ into the frame path to further model the contextual information between different time frames, generating $\boldsymbol{Z}_{b,t} \in \mathbb{R}^{N \times K \times T}$.

The structures are identical in both the frequency path and the frame path, modeling along the frequency dimension and the time dimension respectively. Each path consists of two main modules: the multi-scale selective attention (MSA) module and the full-frequency-frame attention ($\text{F}^3\text{A}$) module. As illustrated in Figure 3, taking the frequency path as an example, we first apply the MSA module along the frequency dimension $K$ to selectively extract features from $\boldsymbol{Z}_b$, which results in enhanced frequency features $\bar{\boldsymbol{Z}}_b \in \mathbb{R}^{N \times K \times T}$. Then, the $\text{F}^3\text{A}$ module is used to integrate information across different sub-bands of $\bar{\boldsymbol{Z}}_b$, followed by layer normalization, to produce the output feature of the frequency path $\boldsymbol{Z}_{b,f}$.

### 3.3.1 MULTI-SCALE SELECTIVE ATTENTION MODULE

The MSA module enhances important features through a selective attention mechanism and is divided into three stages: encoding, fusing, and decoding, as shown in Figure 3(a). Taking the MSA module in the frequency path as an example, the input to the module is $\boldsymbol{Z}_b$.

**The encoding stage**. This stage aims to capture multi-scale acoustic features. Specifically, we first use multiple 1D convolutional layers (with a stride of 2 and channel of $H$) to progressively downsample the frequency dimension to $\frac{K}{2^D}$, resulting in a set of multi-scale acoustic features

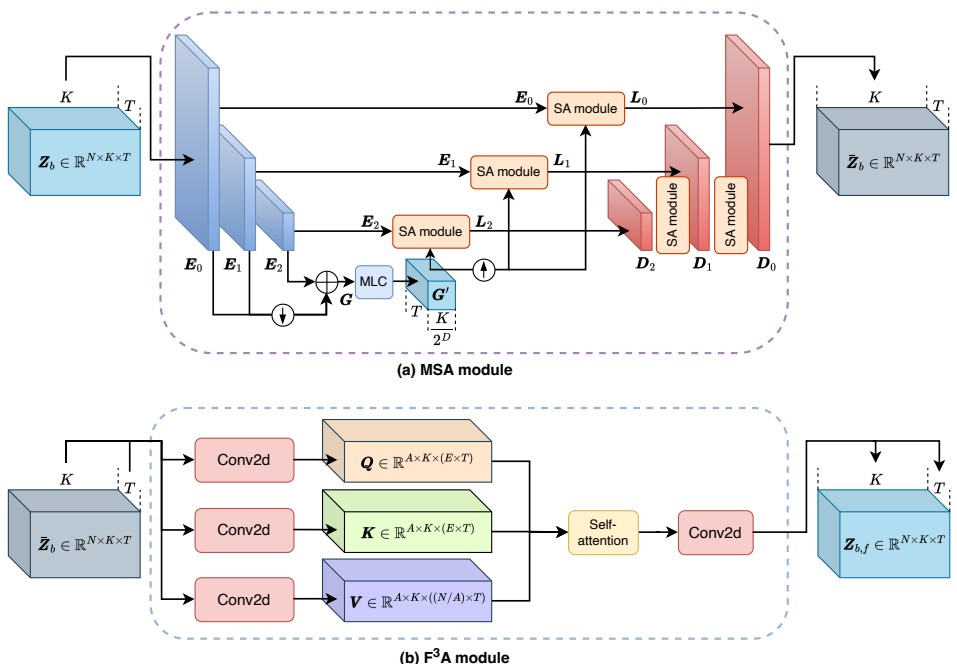

Figure 3: The structure of the MSA module and the $F^3A$ module. The structures of frequency and frame paths are the same.

$\{\boldsymbol{E}_d \in \mathbb{R}^{H \times \frac{K}{2^d} \times T} | d = [0, D]\}$, where $d$ denotes the $d$-th layer of downsampling. Next, we apply average pooling layers, denoted as $\lambda(\cdot)$, to downsample all $\boldsymbol{E}_d$ to the same frequency resolution $\frac{K}{2^D}$. Subsequently, the features with different frequency resolutions are fused into global features $\boldsymbol{G} = \sum_{d=0}^{D} \lambda(\boldsymbol{E}_d), \boldsymbol{G} \in \mathbb{R}^{H \times \frac{K}{2^D} \times T}$ through addition. Finally, a multi-layer convolutional (MLC) network is used to transform $\boldsymbol{G}$ into $\boldsymbol{G}' \in \mathbb{R}^{H \times \frac{K}{2^D} \times T}$.

**The fusing stage**. In this stage, we fuse the local $\boldsymbol{E}_d$ and global $\boldsymbol{G}'$ information using the selective attention (SA) module. Specifically, for the $d$-th layer, we first use two 1D convolutions to map $\boldsymbol{G}'$ into $\tau \in \mathbb{R}^{H \times \frac{K}{2^D} \times T}$ and $\rho \in \mathbb{R}^{H \times \frac{K}{2^D} \times T}$, respectively. Then, we also use one 1D convolution to map $\boldsymbol{E}_d$ into $\phi \in \mathbb{R}^{H \times \frac{K}{2^d} \times T}$. To match the resolution of $\phi$, $\tau$ and $\rho$ are upsampled through interpolation $\mu(\cdot)$, and selective attention weights are generated using a sigmoid function $\sigma(\cdot)$. Finally, the attention weights are multiplied element-wise with $\phi$, and $\mu(\rho)$ is added to the result to obtain $\boldsymbol{L}_d \in \mathbb{R}^{H \times \frac{K}{2^d} \times T}$. The above process can be expressed as follows:

$$\boldsymbol{L}_d = f(\mu(\tau), \phi, \mu(\rho)). \tag{4}$$

The function of $f$ is defined as follows:

$$f(x, y, z) = \sigma(x) \odot y + z, \tag{5}$$

where $x$ and $z$ represent global features, $y$ represents local features, and $\odot$ denotes element-wise multiplication. This function describes the mathematical process of SA mechanism. We first apply sigmoid function to $x$, generating a value between 0 and 1. Then, the value is used to extract effective features from local information by calculating element-wise product of $\sigma(x)$ and $y$. Finally, we add the product to $z$, fusing global information and filtered local information. In this way, $\{\boldsymbol{L}_d \in \mathbb{R}^{H \times \frac{K}{2^d} \times T} \mid d = [0, D]\}$ contains both local and global information, which helps the model better extract the acoustic features in the audio mixture.

**The decoding stage**. In the $d$-th layer, where $d \in [0, D-1]$, the input consists of the decoding result from the previous layer $d + 1$ (denoted as $\boldsymbol{D}_{d+1} \in \mathbb{R}^{H \times \frac{K}{2^{d+1}} \times T}$) and the output $\boldsymbol{L}_d$ from the fusing stage at the $d$-th layer. $\boldsymbol{D}_{d+1}$ is processed through the SA module to produce $\boldsymbol{D}_d$. Specifically, $\boldsymbol{D}_{d+1}$ is transformed using two 1D convolutions to obtain $\alpha \in \mathbb{R}^{H \times \frac{K}{2^{d+1}} \times T}$ and $\beta \in \mathbb{R}^{H \times \frac{K}{2^{d+1}} \times T}$, while $\boldsymbol{L}_d$ is transformed through a 1D convolution to produce $\gamma \in \mathbb{R}^{H \times \frac{K}{2^d} \times T}$. We then compute:

$$\boldsymbol{D}_d = f(\mu(\alpha), \gamma, \mu(\beta)), \tag{6}$$

where $f$ is defined in equation 5. This formulation integrates the decoding result with the output from the fusing stage to generate the next layer of decoded features. In particular, for the layer where $d = D$, $\boldsymbol{D}_D = \boldsymbol{L}_D \in \mathbb{R}^{H \times \frac{K}{2D} \times T}$. For the layer where $d = 0$, we use one 1D convolution to restore the hidden dimension $H$ in $\boldsymbol{D}_0 \in \mathbb{R}^{H \times K \times T}$ to the feature dimension $N$, obtaining $\bar{\boldsymbol{Z}}_b \in \mathbb{R}^{N \times K \times T}$ as the output of the MSA module. In the MSA module of the frequency path, the frequency dimension $K$ is considered the processing dimension. In the frame path, the time dimension $T$ is considered the processing dimension.

### 3.3.2 FULL-FREQUENCY-FRAME ATTENTION MODULE

In the frequency path, the F³A module is used to aggregate features across different sub-bands, as shown in Figure 3(b). Given the input $\bar{\boldsymbol{Z}}_b$ and the number of attention heads $A$, we first use separate $1 \times 1$ 2D convolutional layers with distinct parameters to transform $\bar{\boldsymbol{Z}}_b$ into query $\boldsymbol{Q} \in \mathbb{R}^{(A \times E) \times K \times T}$, key $\boldsymbol{K} \in \mathbb{R}^{(A \times E) \times K \times T}$, and value $\boldsymbol{V} \in \mathbb{R}^{(A \times \frac{N}{A}) \times K \times T}$.

To obtain the information of full time length on each sub-band and apply self-attention mechanism, frame dimension $T$ and the channel dimension $E$ are merged in order of time step, so we get query $\boldsymbol{Q}_i \in \mathbb{R}^{K \times (E \times T)}$ and key $\boldsymbol{K}_i \in \mathbb{R}^{K \times (E \times T)}$ for the $i$-th attention head. Similarly, we get value $\boldsymbol{V}_i \in \mathbb{R}^{K \times (\frac{N}{A} \times T)}$. $\boldsymbol{K}_i$ is transposed and then multiplied with $\boldsymbol{Q}_i$ to calculate the attention map of size $K \times K$, which indicates the similarity between each sub-band and acts as the weight information of the frequency context. Then the attention map is multiplied with $\boldsymbol{V}_i$ to obtain the output matrix. For the $i$-th attention head, the output $\boldsymbol{O}_i \in \mathbb{R}^{K \times (\frac{N}{A} \times T)}$ is calculated as follows:

$$\boldsymbol{O}_i = \text{Softmax} \left( \frac{\boldsymbol{Q}_i \boldsymbol{K}_i^{\text{T}}}{\sqrt{E \times T}} \right) \boldsymbol{V}_i. \tag{7}$$

The output matrix of each attention head is concatenated to get $\boldsymbol{O} \in \mathbb{R}^{K \times (N \times T)}$, and the full-time length is split into $T$ time steps and transformed by 2D convolutional layer, generating the output $\boldsymbol{Z}_{b,f} \in \mathbb{R}^{N \times K \times T}$. The process of the F³A module in the frame path is similar.

### 3.4 BAND RESTORATION MODULE

After going through the separator, the sub-bands need to be converted back to their original width during mask estimation. Specifically, $\boldsymbol{J}_B \in \mathbb{R}^{N \times K \times T}$ denotes the output of the separator. For the $k$-th sub-band feature $\boldsymbol{J}_{B,k} \in \mathbb{R}^{N \times T}$ ($k \in [1, K]$), the PReLU activation function and 1D convolutions are used to transform the number of channels to twice the original dimension $2G_k$, corresponding to the real and the imaginary part. The complex feature is restored to generate a mask for each sub-band $\boldsymbol{M}_k \in \mathbb{C}^{G_k \times T}$ using the ReLU activation function. Then they are merged on the frequency dimension to get the mask for the whole band $\boldsymbol{M} \in \mathbb{C}^{F \times T}$. Similar to band-split, the 1D convolutions of different sub-bands do not share parameters.

## 4 ECHOSET

To develop models that perform better in daily scenarios, we need a dataset close to the real world. We create EchoSet, a speech separation dataset with various noise and realistic reverberation, based on SoundSpaces 2.0 (Chen et al., 2022) and Matterport3D (Chang et al., 2017). An analysis of the dataset is shown in Table 1.

SoundSpaces 2.0 is an audio rendering platform in 3D environments. Given the mesh of a 3D scenario, it can simulate the acoustic effects of any sound captured from microphones. We followed the steps below to generate mixed speech. (1) Choose the scenario. We selected rooms where daily conversations often occur (such as office, living room, bedroom, dining room, etc.) from Matterport3D, a large RGB-D dataset containing 90 diverse multi-floor and multi-room indoor scenes. (2) Define or sample the position. We defined a microphone at a suitable position, like next to a table or sofa, and sampled two sound sources in the same room. (3) Sample the direction and distance. The angle between the microphone and the sound source must be obtuse, meaning that the speaker and listener face each other. The distance between the microphone and each speaker was randomly sampled between 1 m and 5 m. (4) Sample the height. The microphone and sound sources were randomly generated at a vertical height of 1.5 m to 1.9 m from the floor, which is about a person's

height. (5) Generate the audio. With SoundSpaces 2.0, mixed audio files were generated based on bidirectional path tracking algorithm (Cao et al., 2016), which can simulate various effects in the sound propagation process, including reverberation, diffraction, and absorption. Materials of the room wall and the objects were annotated by Matterport3D and considered during the generation of the audio mixture.

Based on the SoundSpaces 2.0 platform and the Matterport 3D scene dataset, we can simulate reverberant audio from different speakers in LibriSpeech (Panayotov et al., 2015) to build a new dataset, EchoSet. In total, EchoSet includes 20,268 training utterances, 4,604 validation utterances, and 2,650 test utterances. Each utterance lasts for 6 seconds. We mixed the speech of the two speakers at a random overlap ratio and added some noises from WHAM! noise (Wichern et al., 2019). The two different speakers were mixed with signal-to-distortion ratio (SDR) sampled between -5 dB and 5 dB. The noises were mixed with SDR sampled between -10 dB and 10 dB. The dataset is available at: `https://huggingface.co/datasets/JusperLee/EchoSet`.

| Dataset | Noise | Reverb | Overlapping |
|---|---|---|---|
| WSJ0-2mix (Hershey et al., 2016) | × | × | Full |
| WHAM! (Wichern et al., 2019) | √ | × | Full |
| WHAMR! (Maciejewski et al., 2020) | √ | Only room | Full |
| Libri2Mix (Cosentino et al., 2020) | √ | × | Sparse but fixed |
| LRS2-2Mix (Li et al., 2023) | √ | Room and objects (different scenes) | Full |
| EchoSet (*ours*) | √ | Room and objects (same scene) | Random |

Table 1: Features of datasets for speech separation.

## 5 EXPERIMENTAL SETUP

**Dataset**. We report the performance of TIGER on EchoSet. For fair comparison with previous speech separation methods (Li et al., 2023; Wang et al., 2023; Hu et al., 2021), we also used two benchmark datasets LRS2-2Mix (Li et al., 2023) and Libri2Mix `train-100` *min* (Cosentino et al., 2020). All of these datasets are at a sampling rate of 16 kHz.

To validate the gap between EchoSet and real-world environments, we constructed real-world data by selecting 10 real-world environments and recording audio from 40 speakers from the LibriSpeech (Panayotov et al., 2015) test set. The two audio used for mixing were recorded in the same acoustic scene (e.g., the shape and material of the walls and objects in the room) and followed the same mixing method as LRS2-2Mix (Li et al., 2023). The duration of each audio is 60 seconds, and the sampling rate is 16 kHz. For more details of these datasets, please refer to Appendix A.

**Training and evaluation**. During training, we utilized 3-second audio segments for EchoSet and Libri2Mix, and 2-second segments for LRS2-2Mix. The negative SI-SDR was adopted as the training loss (Le Roux et al., 2019). Adam optimizer (Kingma & Ba, 2014) was employed with an initial learning rate of 0.001, adjusted based on validation performance. Evaluation metrics included SDRi and SI-SDRi (Vincent et al., 2006), with higher values indicating better performance. We report parameters and MAC operations for complexity, which are calculated for one second of audio at 16 kHz. Inference speed was measured on NVIDIA RTX 4090 and Intel Xeon Gold 6326. Detailed training and evaluation configurations can be found in Appendix B and Appendix C. Code is available at: `https://github.com/JusperLee/TIGER`.

## 6 RESULTS AND DISCUSSION

### 6.1 ECHOSET IS MORE CLOSE TO THE REAL-WORLD DATA

We trained different models on Libri2Mix, LRS2-2Mix and EchoSet, and then tested them on the data collected in the real world. The results are presented in Figure 4. Compared to models trained on Libri2Mix and LRS2-2Mix, the models trained on EchoSet produced higher-quality separated speech, confirming that the gap between EchoSet and real-world audio is relatively small.

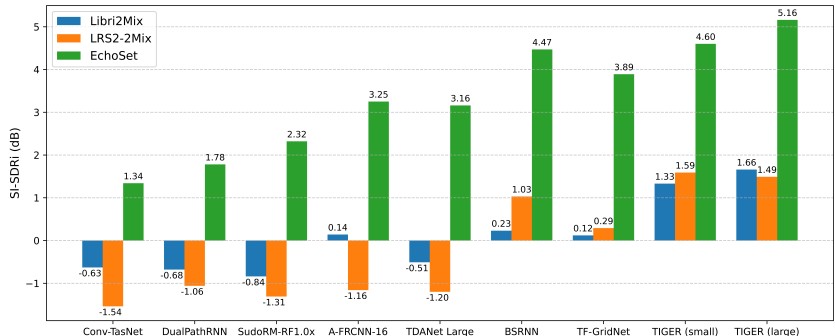

Figure 4: SI-SDRi results of different models on the real-world data. Models were trained on *Libri2Mix, LRS2-2Mix and EchoSet* respectively.

| Methods | Libri2Mix | | LRS2-2Mix | | EchoSet | |
|---|---|---|---|---|---|---|
| | SDRi | SI-SDRi | SDRi | SI-SDRi | SDRi | SI-SDRi |
| Conv-TasNet (Luo & Mesgarani, 2019) | 12.50 | 12.10 | 11.0 | 10.6 | 7.69 | 6.89 |
| DualPathRNN (Luo et al., 2020) | 11.64 | 11.26 | 13.0 | 12.7 | 5.87 | 5.06 |
| SudoRM-RF1.0x (Tzinis et al., 2020) | 13.58 | 13.16 | 11.4 | 11.0 | 7.70 | 6.84 |
| A-FRCNN-16 (Hu et al., 2021) | 16.73 | 16.32 | 13.3 | 13.0 | 9.64 | 8.76 |
| TDANet Large (Li et al., 2023) | 16.11 | 15.64 | 14.5 | 14.2 | 10.14 | 9.21 |
| BSRNN (Luo & Yu, 2023) | 17.38 | 16.96 | 14.4 | 14.1 | 12.75 | 12.23 |
| TF-GridNet (Wang et al., 2023) | **19.56** | **19.24** | **15.7** | **15.4** | 13.73 | 12.85 |
| TIGER (small) | 17.09 | 16.67 | 14.2 | 13.9 | 13.15 | 12.58 |
| TIGER (large) | 18.34 | 17.97 | 15.3 | 15.1 | **14.22** | **13.73** |

Table 2: Performance comparison of TIGER and other separation models on Libri2Mix, LRS2-2Mix, and EchoSet. *Models are trained and tested on corresponding datasets.* Bold denotes the best performance, and underline indicates the second-best. SDRi and SI-SDRi are recorded in dB.

## 6.2 COMPARISONS WITH STATE-OF-THE-ART METHODS

We compared TIGER with previous SOTA models including Conv-TasNet (Luo & Mesgarani, 2019), DualPathRNN (Luo et al., 2020), SudoRM-RF1.0x (Tzinis et al., 2020), A-FRCNN-16 (Hu et al., 2021), TDANet Large (Li et al., 2023), BSRNN (Luo & Yu, 2023) and TF-GridNet (Wang et al., 2023) in terms of performance and efficiency. TIGER (small) and TIGER (large) denote the models with the number of FFI blocks $B = 4$ and $B = 8$, respectively.

**Separation performance**. TIGER obtained competitive separation performance on the three datasets compared with previous SOTA models (see Table 2). On Libri2Mix, which is relatively simple for lack of noise and reverberation, TIGER (large) was second only to the current SOTA model TF-GridNet, with a 6% drop in performance. On LRS2-2Mix, a more complicated dataset with reverberation recorded in different scenes, the drop in performance of TIGER (large) was only 2% compared with TF-GridNet. On EchoSet, the only dataset with the most realistic reverberation among the three, TIGER (large) achieved an SDRi of 14.22 dB, surpassing other existing methods. On this dataset, TIGER (small) also achieved the performance that was only slightly lower than TF-GridNet. From the above experimental results, we can see that the more complex the acoustic scenarios are, the better performance TIGER will produce. Similarly, based on the results in Figure 4, we observed that TIGER also outperforms existing models in real-world test scenarios. This demonstrates that TIGER is applicable to complex real-world acoustic scenarios including diverse noise and reverberation. To visualize the separation result, we present the spectrogram differences between the audio separated by TIGER and TF-GridNet (Appendix I), demonstrating that TIGER is capable of effectively reconstructing both low-frequency and high-frequency features.

TIGER also demonstrates advanced performance on cinematic sound separation, which aims to extract different audio elements from a film's soundtrack. See Appendix D for details. As for demos of speech and cinematic sound separation, please refer to the project page[1].

---

[1] https://cslikai.cn/TIGER

| Methods | Paras (M) | MACs (G/s) | Training | | Inference | | |
|---|---|---|---|---|---|---|---|
| | | | GPU Time | GPU Memory | CPU Time | GPU Time | GPU Memory |
| Conv-TasNet 2019 | 5.62 | 7.19 | 92.96 | 1436.94 | **64.21** | **13.17** | 28.78 |
| DualPathRNN 2020 | 2.72 | 45.01 | **67.23** | 1813.55 | 723.13 | 30.38 | 298.09 |
| SudoRM-RF1.0x 2020 | 2.72 | **4.65** | 118.46 | 1353.43 | 104.32 | 20.66 | **24.42** |
| A-FRCNN-16 2021 | 6.13 | 81.28 | 230.53 | 2925.83 | 478.58 | 82.65 | 163.82 |
| TDANet Large 2023 | 2.33 | 9.19 | 263.43 | 4260.36 | 434.44 | 74.27 | 136.96 |
| BSRNN 2023 | 25.97 | 98.70 | 258.55 | **1093.11** | 897.27 | 78.27 | 130.24 |
| TF-GridNet 2023 | 14.43 | 323.75 | 284.17 | 5432.94 | 2019.60 | 94.30 | 491.73 |
| TIGER (small) | **0.82** | 7.65 | 160.17 | 2049.46 | 351.15 | 42.38 | 122.23 |
| TIGER (large) | **0.82** | 15.27 | 229.23 | 3989.59 | 765.47 | 74.51 | 122.23 |

Table 3: Efficiency comparisons of TIGER and other models. GPU Time and CPU Time are recorded in ms, and GPU Memory is recorded in MB.

**Separation effciency**. The parameters of TIGER were only 0.82 M, and the MACs were only 7.65 G/s and 15.27 G/s for the small and large versions respectively. Compared with TF-GridNet, the parameters of TIGER (large) dropped by 94.3%, and the MACs were reduced by 95.3%. For inference of one-second audio, TIGER (large) took about 1/3 of the CPU Time and 3/4 of the GPU Time compared with TF-GridNet, demonstrating a significant calculation compression effect. Besides, TIGER took up less memory during training and inference, making TIGER more suitable for deployment on devices with limited computational resources.

## 6.3 ABLATION STUDY

We adopted the small version of TIGER ($B = 4$) in the ablation studies. All the models were trained and tested on *EchoSet*. The training configuration of TIGER and other models was the same.

| Schemes | Sub-band Number | SDRi (dB) | SI-SDRi (dB) | Paras (M) | MACs (G/s) | GPU Time (ms) |
|---|---|---|---|---|---|---|
| NonSplit | 321 | 11.53 | 11.12 | **0.56** | 40.89 | 72.91 |
| NormalSplit | 47 | 12.94 | 12.37 | 0.82 | **5.29** | **40.78** |
| EvenSplit | 67 | 12.80 | 12.24 | 0.82 | 7.65 | 43.06 |
| LowFreqNarrowSplit (*ours*) | 67 | **13.15** | **12.58** | 0.82 | 7.65 | 42.38 |

Table 4: Comparison of performance and efficiency of models with different band-split schemes.

| MSA module | F$^3$A module | SDRi (dB) | SI-SDRi (dB) | Params (M) | MACs (G/s) |
|---|---|---|---|---|---|
| × | √ | 7.58 | 7.22 | **0.33** | **2.74** |
| √ | × | 12.34 | 11.87 | 0.75 | 4.95 |
| √ | √ | **13.15** | **12.58** | 0.82 | 7.65 |

Table 5: Importance of MSA and F$^3$A modules on the EchoSet test set.

**Band-split schemes.** To verify the effectiveness of the band-split method on the speech separation task, we designed several experiments of different band-split schemes (see Table 10 in Appendix). For these experiments, we kept the feature channel $N$ the same. For the model NonSplit that did not adopt band-split, each frequency point was treated as a sub-band and the real and imaginary channels were transformed to the feature dimension $N$. For the other models, the same method as TIGER was used. A detailed description of the different band-split schemes can be found in Appendix E.

Table 4 shows the performance and efficiency of models using different band-split schemes. While adopting band-split increased the number of parameters due to non-shared 1D convolutions, it significantly reduced overall computational costs by decreasing the total number of sub-bands. This approach allowed the model to focus on important frequency bands, low and medium bands for speech since human speech typically ranges from 85 Hz to 1100 Hz (Loizou, 1998). The LowFreq-NarrowSplit scheme offered finer splits in low-frequency bands compared to NormalSplit, resulting in enhanced performance. In contrast, EvenSplit maintained the same number of sub-bands with an even distribution, leading to a drop in SDRi and SI-SDRi compared to LowFreqNarrowSplit, which highlights the effectiveness of band-split in capturing critical frequency information.

| Replacement structures | SDRi (dB) | SI-SDRi (dB) | Params (M) | MACs (G/s) | GPU Time (ms) |
|:---:|:---:|:---:|:---:|:---:|:---:|
| LSTM | **13.92** | **13.41** | 2.05 | 49.38 | 83.16 |
| Mamba | 13.02 | 12.59 | 1.33 | 21.03 | 59.91 |
| SRU | 12.87 | 12.43 | 1.00 | 20.30 | 53.26 |
| MSA (*ours*) | 13.15 | 12.58 | **0.82** | **7.65** | **42.38** |

Table 6: Comparison of performance with different structures to replace MSA module.

| Replacement structures | SDRi (dB) | SI-SDRi (dB) | Params (M) | MACs (G/s) |
|:---:|:---:|:---:|:---:|:---:|
| LSTM | 12.64 | 12.05 | 2.46 | 51.59 |
| Mamba | 12.20 | 11.78 | 1.74 | 25.43 |
| SRU | 12.48 | 11.97 | 1.41 | 22.71 |
| $F^3A$ (*ours*) | **13.15** | **12.58** | **0.82** | **7.65** |

Table 7: Comparison of performance with different structures to replace $F^3A$ module.

**Importance of MSA and $F^3A$ modules**. We investigated the role of the MSA and $F^3A$ modules in model performance. To this end, we constructed two controlled models, removing each of these modules. As shown in Table 5, removing the MSA module resulted in the worst results, which validates the effectiveness of the MSA module in speech separation, as it fully integrates multi-scale features. The performance also decreased after the $F^3A$ module was removed, indicating that the global integration of time and frequency helps TIGER extract relevant auditory features. Overall, the results show the MSA and $F^3A$ modules play an important role in improving performance.

Furthermore, MSA module and $F^3A$ module can be replaced by other structures for sequence data modeling, such as LSTM (Graves & Graves, 2012), SRU (Lei et al., 2018), and Mamba (Gu & Dao, 2023). We then evaluated the impact of replacing the MSA and $F^3A$ modules with different sequence modeling structures. We first replaced the MSA module in TIGER with LSTM, SRU, and Mamba, with detailed replacement methods provided in Appendix F. The experimental results are shown in Table 6. We observed that the MSA module significantly reduced the computational load while maintaining strong performance. Although LSTM demonstrated better performance in sequence data modeling, the iterative nature of RNN computations resulted in the GPU inference time being twice as long as that of the MSA-based separator. While linear RNN structures like SRU and Mamba sped up inference to some extent, there remained a gap in separation performance and efficiency compared to the MSA module. This highlights the importance of leveraging multi-scale information for both temporal and frequency modeling.

Next, we replaced the self-attention mechanism in the $F^3A$ module with LSTM, SRU, and Mamba to evaluate the effect of different structural replacements. The experimental results are presented in Table 7. We found that the $F^3A$ module produced the best results among the four experiments, mainly because long-range dependencies captured by the self-attention module help enhance the global context of frequency and temporal features.

We also verified the impact of alternating the frequency path and frame path on model performance, and explored the lightweight potential of TIGER under a smaller parameter scale. See Appendix G and H for more details.

## 7 CONCLUSION

In this paper, we present TIGER, an efficient time-frequency domain speech separation model with significantly reduced parameters and computational costs. TIGER effectively extracts key acoustic features through frequency band-split, multi-scale and full-frequency-frame modeling. We also introduce the EchoSet dataset that simulates realistic acoustic scenarios. Experiments showed that TIGER outperformed existing SOTA models in complex acoustic environments, with 94.3% fewer parameters and 95.3% less computational costs, and demonstrated good generalization ability in the task of movie audio separation. TIGER provides new ideas for designing lightweight speech separation models suitable for devices with limited resources.

ACKNOWLEDGMENTS

This work was supported in part by the National Key Research and Development Program of China (No. 2021ZD0200301) and the National Natural Science Foundation of China (No. U2341228).

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

## A DATASET DETAILS

**EchoSet**. This dataset includes 20268 training utterances, 4604 validation and 2650 test utterances. The length of each audio is 6 seconds. The target speech was selected from LibriSpeech (Panayotov et al., 2015), mixed with SDR ranging from -5 dB to 5 dB. The speech and noise which was sampled from WHAM! were mixed with SDR sampled between -10 dB and 10 dB. This dataset contains realistic reverberation. The sampling rate is 16 kHz.

**LRS2-2Mix** (Li et al., 2023). Each audio in this dataset lasts for 2 seconds, at the sampling rate of 16 kHz. The training set, validation set and test set are about 11.1, 2.8 and 1.7 hours, respectively. The utterances were selected from the LRS2 (Afouras et al., 2018) corpus, which consists of video clips acquired through BBC, and were mixed with SDR sampled between -5 dB and 5 dB. Since the audio files were recorded in real acoustic scenarios, LRS2-2Mix contains much noise and reverberation.

**Libri2Mix** (Cosentino et al., 2020). Each audio in this dataset lasts for 3 seconds. The target speech for each audio mixture was randomly chosen from LibriSpeech (Panayotov et al., 2015) (*train-100*) and combined with a uniformly sampled Loudness Units relative to Full Scale (Series, 2011) between -25 and -33 dB. We adoped the 16 kHz *min* version with no noise or reverberation in our experiments.

**Real-world data**. We collected a small-scale dataset from the physical world to test the performance of models trained on different datasets in real-world scenarios, with each audio clip 60 seconds long. Its data collection process is described as follows. First, we selected 10 rooms of varying sizes and shapes as distinct acoustic environments. Then, we randomly sampled approximately 1.5 hours of 16 kHz speech audio from the LibriSpeech test set (Panayotov et al., 2015), and sampled noise data from the WHAM! noise dataset (Wichern et al., 2019). During the recording process, audio content was played using the speakers of a 2023 MacBook Pro and recorded via a Logitech Blue Yeti Nano omnidirectional microphone placed in a fixed position. The distance between the speaker and the microphone was randomly selected from 0.3 m to 2 m. The recording parameters were set to a 16 kHz sampling rate and 32-bit depth. This setup ensured that both speech and noise were recorded in the same room, preserving the authenticity of the reverberation effects. Finally, we processed the collected audio by mixing the recordings. Specifically, audio from different speakers was mixed using SDRs randomly sampled between -5 dB and 5 dB. Noise data was added using SDRs randomly sampled between -10 dB and 10 dB. Since the propagation paths of sounds in the air are independent of one another, mixing these components is considered a reasonable approach. This design ensures the realism and diversity of the evaluation dataset, effectively capturing the complexity of speech separation in real-world conditions.

## B TRAINING CONFIGURATION

In the encoder and decoder, the window and hop size of STFT and iSTFT were set to 640 (40 ms) and 160 (10 ms). We use the Hanning window to mitigate spectrum leakage. According to the

Nyquist sampling theorem, the frequency range represented was 0-8 kHz for audio with a sampling rate of 16 kHz. In this way, each frame was represented by 321-dimensional complex spectra, and the frequency resolution was 25 Hz. We adopt the band-split scheme LowFreqNarrowSplit in Table 10. The number of total sub-bands $K$ was 67. For each sub-band, the bandwidth was uniformly transformed into $N = 128$. In the separator, the FFI blocks which share parameters were repeated $B = 4$ times for the small version and $B = 8$ times for the large version. Each MSA module's features were downsampled for $D = 4$ times, and the hidden layer dimension $H$ was set to 256. For F$^3$A module, the number of attention heads was set to 4. When calculating the query and key in each head of the F$^3$A module, the hidden channel $E$ was set to 4.

During training, We used a 3-second audio segment for EchoSet and Libri2Mix, and a 2-second for LRS2-2Mix. We used the maximization of SI-SDR as the training loss (Le Roux et al., 2019). The maximum training round was 500. We used Adam as the optimizer (Kingma & Ba, 2014), with the initial learning rate set to 0.001. If the loss on the validation set did not decrease further within 10 consecutive rounds, the learning rate was halved. When the performance on the validation set did not improve further within 20 consecutive rounds, the training was stopped.

## C    EVALUATION CONFIGURATION

In all experiments, we reported the quality of separated audio on SDRi (Vincent et al., 2006) and SI-SDRi (Le Roux et al., 2019):

$$\text{SDRi} = \text{SDR}(\bar{\boldsymbol{P}}_i, \boldsymbol{P}_i) - \text{SDR}(\boldsymbol{S}, \boldsymbol{P}_i), \tag{8}$$

$$\text{SI-SDRi} = \text{SI-SDR}(\bar{\boldsymbol{P}}_i, \boldsymbol{P}_i) - \text{SI-SDR}(\boldsymbol{S}, \boldsymbol{P}_i), \tag{9}$$

When evaluating model performance on real-world data, we used the training lengths (Libri2Mix: 3s, LRS2-2Mix: 2s, EchoSet: 3s) of the respective datasets to inference the 60-second audio with a 50% overlap sliding scale. This approach to some extent mitigates the problem of model performance degradation that may be caused by the difference in training and inference lengths, thus ensuring fairness in the model's performance comparison on the real-world data.

To measure the complexity of the model, we used parameters and multiply-accumulate operations (MACs) for theoretical analysis. In the speech separation task, since the audio length is not fixed, we used MACs for separating one-second audio as an indicator for complexity evaluation. We used ptflops 0.7.3[2] to calculate parameters and MACs. For actual evaluation, we performed the backward process (training) and forward process (inference) 1000 times, respectively, on one second of audio at a 16 kHz sampling rate, then took the average to indicate the training and inference speed. We reported the GPU time and GPU memory usage during the training process, as well as the CPU time, GPU time, and GPU memory usage during the inference process. To simulate the limited computational conditions of mobile devices on which the speech separation model is deployed in real-world situations, we fixed the number of threads to 1 when calculating CPU (Intel(R) Xeon(R) Gold 6326) time and only used a single card when calculating GPU (GeForce RTX 4090) time.

## D    CINEMATIC SOUND SEPARATION TASK

The cinematic sound separation task (Uhlich et al., 2024) is to separate different signals from mixed audio, including speech, music and sound effects. We migrated TIGER to cinematic sound separation to test the generalization ability of the model on similar tasks.

We tested TIGER's performance on the DnR dataset, which consists of three tracks: speech, music, and sound effects. The length of each audio is 60 seconds. Each track does not completely overlap, and the sampling rate is 44.1 kHz. The dataset is composed of 3295 training audio, 440 validation audio, and 652 test audio.

According to the composition of the mixed audio, the band-split scheme was adjusted as shown in Table 8. Since the frequency of human hearing ranges from 20 Hz to 20000 Hz, there was no need to split the high-frequency band above 20000 Hz. The window size $W$ of STFT was set to 2048,

---

[2]https://pypi.org/project/ptflops/0.7.3/

| Range (Hz) | Width (Hz) | Total sub-band number |
|---|---|---|
| 0-1000 | 50 | 20 |
| 1000-2000 | 100 | 10 |
| 2000-4000 | 250 | 8 |
| 4000-8000 | 500 | 8 |
| 8000-16000 | 1000 | 8 |
| 16000-20000 | 2000 | 2 |
| 20000-44100 | 22100 | 1 |

Table 8: Band-split scheme on DnR

and the stride $J$ was set to 512. The feature dimension was set to $N = 132$. In the separator, the FFI blocks were repeated for $B = 8$ times. Other settings remained unchanged.

As for the training configuration, in order to improve the speed of the training phase, each 60-second training audio in DnR was segmented using Voice Activity Detection (VAD). Then 3 seconds of audio was randomly sampled from each component to synthesize the mixed audio. The sum of the mean absolute error (MAE) in the frequency domain and the time domain was used as the training loss, which was the same as (Uhlich et al., 2024):

$$\mathcal{L} = \frac{1}{C} \sum_{i=1}^{C=3} |\bar{\boldsymbol{P}}_i - \boldsymbol{P}_i| + \frac{1}{C} \sum_{i=1}^{C=3} |\text{STFT}(\bar{\boldsymbol{P}}_i) - \text{STFT}(\boldsymbol{P}_i)|. \tag{10}$$

The maximum training epochs were 500. AdamW was used as the optimizer, and the initial learning rate was set to $lr = 0.001$. If the loss on the validation set did not decrease further within 5 consecutive rounds, the learning rate was reduced by half. When the performance on the validation set did not improve further within 10 consecutive rounds, the training process was stopped.

During inference, we employed a sliding window approach, dividing the 60-second audio into 6-second overlapping segments with a 50% overlap, and then reassembling the segments back to their original length after processing.

| Structures | Music (dB) | Speech (dB) | Sound Effect (dB) | Paras (M) | MACs (G/s) |
|---|---|---|---|---|---|
| Conv-TasNet* | 0.3 | 8.5 | 2.0 | 5.3 | 19.82 |
| MRX* | 4.2 | 12.3 | 5.7 | 30.51 | 10.59 |
| BSRNN | 5.5 | 13.8 | 1.8 | 52.8 | 18.2 |
| TIGER | **7.4** | **15.5** | **6.5** | **1.40** | **4.07** |

Table 9: Comparison of performance and efficiency of cinematic sound separation models on DnR. '*' means the result comes from the original paper of DnR (Petermann et al., 2022).

The experimental results are shown in Table 9. TIGER demonstrated outstanding reconstruction performance across the three audio tracks. Specifically, for the tasks of separating music, speech, and sound effects, TIGER achieved SI-SDR scores of 7.4 dB, 15.5 dB, and 6.5 dB, respectively, significantly outperforming BSRNN. This indicates that TIGER has a stronger capacity for capturing audio features.

Moreover, TIGER's parameters were only 1.40 million, and its computational costs were 4.07 G MACs per second, which kept resource usage at a low level and was very efficient. These results further validated TIGER's effectiveness in the domain of cinematic sound separation, providing a strong foundation for practical applications.

## E  DETAILS OF DIFFERENT BAND-SPLIT SCHEMES

In Table 10, we list several band-split schemes. For datasets of 16 kHz, the full band ranges from 0-8 kHz. Because real-to-complex STFT satisfies the conjugate symmetry, the result can be expressed

| Scheme | Range (Hz) | Width (Hz) | Number | Total number |
|---|---|---|---|---|
| NonSplit | 0-8000 | 25 | 321 | 321 |
| NormalSplit | 0-1000 | 50 | 20 | |
| | 1000-2000 | 100 | 10 | |
| | 2000-4000 | 250 | 8 | 47 |
| | 4000-8000 | 500 | 8 | |
| | 8000 | - | 1 | |
| LowFreqNarrowSplit | 0-1000 | 25 | 40 | |
| | 1000-2000 | 100 | 10 | |
| | 2000-4000 | 250 | 8 | 67 |
| | 4000-8000 | 500 | 8 | |
| | 8000 | - | 1 | |
| EvenSplit | 0-6600 | 100 | 66 | 67 |
| | 6600-8000 | 1400 | 1 | |

Table 10: Different frequency band-split schemes and their corresponding frequency ranges, bandwidths, and numbers of sub-bands.

using only one side. According to the implementation of the *torch.stft*[3], when the window size was set to 640, the encoding dimension was $\lfloor 640/2 \rfloor + 1 = 321$.

For the NonSplit scheme, we didn't apply band-split and kept the original frequency samples 321. The width of each sub-band was 25 Hz. The total sub-band number was 321. We write the mixed audio after STFT as $\boldsymbol{X} \in \mathbb{C}^{F \times T}$. The real and imaginary part of $\boldsymbol{X}$ were treated as two channels and stacked on the channel dimension to obtain feature $\dot{\boldsymbol{X}} \in \mathbb{R}^{2 \times F \times T}$. Then a 2D convolutional layer was applied to $\dot{\boldsymbol{X}}$ to expand the channel dimension to $N$. In this way, we got the input $\boldsymbol{Z} \in \mathbb{R}^{N \times K \times T}$ for the separator ($K = F = 321$ in this case).

For the NormalSplit scheme, we split finer in the low-frequency part. Specifically, we split 0-1000 Hz by a 50 Hz bandwidth. Since the resolution was 25Hz, 2 frequency samples were treated as one band. The total sub-band number in 0-1000 Hz was 20. Accordingly, $G_k = 2$ when $k \in [1, 20]$. Similarly, we split 1000-2000 Hz by a 100 Hz bandwidth. 4 frequency samples were treated as one sub-band and the total sub-band number in 1000-2000 Hz was 10, i.e. $G_k = 4$ when $k \in [21, 30]$. For 2000-4000 Hz and 4000-8000 Hz, 10 frequency samples and 20 frequency samples were treated as one band, respectively. Therefore $G_k = 10$ when $k \in [31, 38]$ and $G_k = 20$ when $k \in [39, 46]$. Since there were 321 frequency points in total, there was one endpoint left, corresponding to 8000 Hz. Thus $G_k = 1$ when $k = 47$. There were 47 sub-bands in total. When adopting band-split strategy, the real part $\text{Re}(\cdot)$ and imaginary part $\text{Im}(\cdot)$ of the frequency sub-band $\boldsymbol{B}_k$ are no longer treated as two channels, but are merged into the frequency dimension. Then we obtain $\dot{\boldsymbol{B}}_k \in \mathbb{R}^{2G_k \times T}$. Group normalization layers and 1D convolutions are used to map the frequency dimension $2G_k$ to the feature dimension $N$, and then $K$ sub-bands are stacked to obtain the input feature $\boldsymbol{Z} \in \mathbb{R}^{N \times K \times T}$ for the separator.

For the LowFreqNarrowSplit scheme, we split the low-frequency area less roughly. In the range of 0-1000 Hz, we split the band by 25 Hz for each sub-band. This way, 1 frequency sample was treated as a sub-band, and the total sub-band number in 0-1000 Hz was 40. Other bands remained the same as NormalSplit. Therefore, we had $G_k = 1$ when $k \in [1, 40]$; $G_k = 4$ when $k \in [41, 50]$; $G_k = 10$ when $k \in [51, 58]$; $G_k = 20$ when $k \in [59, 66]$; $G_k = 1$ when $k = 67$. The implementation kept the same as the NormalSplit scheme.

For EvenSplit, 0-6600 Hz was split evenly by 100 Hz sub-bands. Each sub-band consisted of 4 frequency samples. The remaining part was treated as one sub-band. Accordingly, we had $G_k = 4$ when $k \in [1, 66]$; $G_k = 57$ when $k = 67$. The band-split detail was also the same as the NormalSplit scheme.

---

[3]https://pytorch.org/docs/stable/generated/torch.stft.html

## F  DIFFERENT STRUCTURES IN MSA AND F³A MODULES

In the experiments where we replaced the MSA and F³A modules, we used LSTM (Graves & Graves, 2012), SRU (Lei et al., 2018), and Mamba (Gu & Dao, 2023) as the alternative model structures. When we substituted LSTM for the MSA module, the input $Z_b \in \mathbb{R}^{N \times K \times T}$ is first normalized by group normalization. Then we apply a bi-directional LSTM with the hidden size the same as the hidden layer dimension $H$ in the MSA module, generating the hidden feature $Z'_b \in \mathbb{R}^{2H \times K \times T}$. Next we restore the hidden layer dimension to the input dimension using linear projection. The output of LSTM is $\bar{Z}_b \in \mathbb{R}^{N \times K \times T}$. The configuration for SRU was consistent with that of LSTM. For Mamba, since it is a causal model and cannot access future information, we utilized the BMamba layer, as proposed in SPMamba (Li et al., 2024), to model sequence information bidirectionally, followed by a linear layer to compress the feature channels.

## G  ABLATION STUDY: TIME-FREQUENCY INTERLEAVING

| Structure | SDRi (dB) | SI-SDRi (dB) | Paras (M) | MACs (G/s) | GPU Time (ms) |
|---|---|---|---|---|---|
| T-T | 12.91 | 12.32 | 0.82 | 7.75 | 42.72 |
| F-F | 10.57 | 9.69 | 0.82 | **7.53** | **41.39** |
| F-T (ours) | **13.15** | **12.58** | **0.82** | 7.65 | 42.38 |

Table 11: Comparison of performance and efficiency of models with different modeling paths in the FFI block. T-T means the FFI block consists of two frame paths, while F-F means the FFI block consists of two frequency paths.

In the separator of TIGER, we model time and frequency features of the mixed audio alternately. To demonstrate the effect of time-frequency interleaved structure, we tested the performance of F-F and T-T structures. For F-F, we replace the frame path with the frequency path in the FFI blocks. In other words, each FFI block only includes two frequency paths which process the input $Z_b \in \mathbb{R}^{N \times K \times T}$ and $Z_{b,f} \in \mathbb{R}^{N \times K \times T}$ in the same way but don't share parameters. All FFI blocks still share parameters. The implementation is similar for T-T.

According to the result shown in Table 11, compared with only modeling time or only modeling frequency, the time-frequency interleaved structure can better capture time and frequency information of audio, which facilitates improving performance while keeping the model lightweight.

## H  THE LIGHTWEIGHT POTENTIAL OF TIGER

To further illustrate the lightweight potential of TIGER, we present the experimental results of a smaller version of TIGER as well as the compressed SudoRM-RF model (Tzinis et al., 2020) based on GC3 method (Luo et al., 2021) on EchoSet. The hyperparameters of TIGER (tiny) were set as follows: the feature dimension $N$ was reduced from 128 to 24; the hidden layer dimension $H$ was reduced from 256 to 64; the number of FFI blocks $B$ was 4. The hyperparameter settings for SudoRM-RF were strictly in accordance with the publicly available configurations of GC3[4]. The efficiency was evaluated on one-second audio input. As shown in Table 12, TIGER (tiny) significantly outperforms SudoRM-RF + GC3 at comparable efficiencies, which proves that TIGER is a very effective lightweight model.

## I  VISUALIZATION

In order to intuitively demonstrate the separation performance of TIGER, we provide some examples for visualization, as shown in Figure 5. The following spectrograms show the inference results of TIGER (large) and TF-GridNet on the same audio, and the ground truth. Sample I and II show

---

[4]`https://github.com/yluo42/GC3`

| Method | SDRi (dB) | SI-SDRi (dB) | Paras (K) | MACs (G/s) | GPU Time (ms) |
|---|---|---|---|---|---|
| SudoRM-RF + GC3 | 5.0 | 4.6 | 303.57 | 0.81 | 32.68 |
| TIGER (tiny) | **10.7** | **10.4** | **102.12** | **0.72** | **30.21** |

Table 12: Performance and efficiency comparison of SudoRM-RF + GC3 and TIGER (tiny).

that TIGER produces finer reconstruction results at high frequencies compared with TF-GridNet. TIGER also has better effects in noise reduction and spectrum leakage prevention, as illustrated in Sample III and IV.

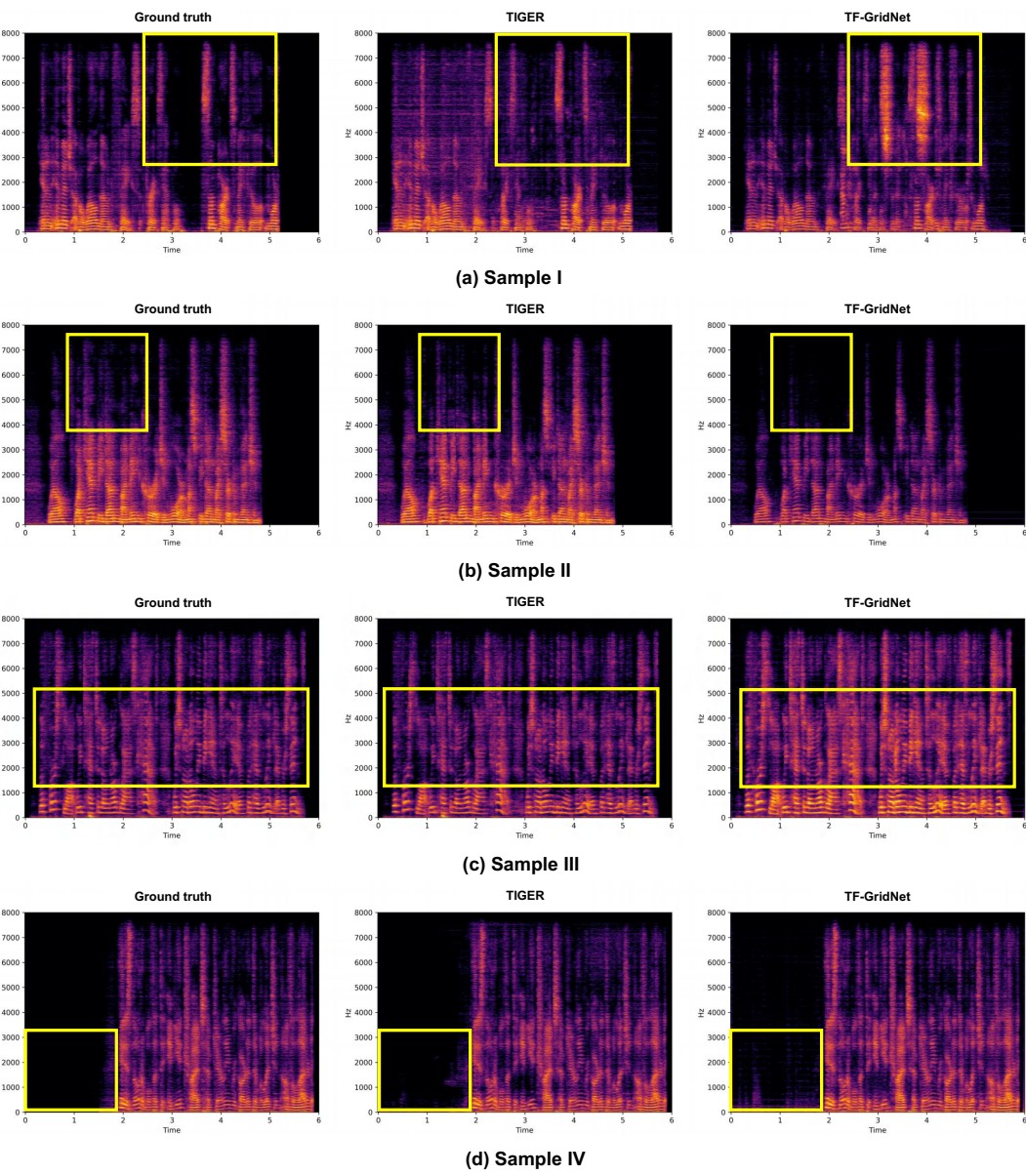

Figure 5: Comparison of the spectrograms of the ground truth, audio separated by TIGER and by TF-GridNet.

