# OpenReview forum: "TIGER: Time-frequency Interleaved Gain Extraction and Reconstruction for Efficient Speech Separation"
_ICLR.cc/2025/Conference — ICLR 2025 Poster_

### Official Review · Reviewer_awAb · 2024-10-26

**Soundness:** 1
**Presentation:** 3
**Contribution:** 1
**Rating:** 1
**Confidence:** 5

**Summary:**

The paper has two main contributions:
1. The TIGER architecture which is a lightweight time-frequency model
focusing on reducing model size while keeping separation accuracy high.
2. The EchoSet dataset for two speaker speech separation which attempts
to resemble real world scenarios more closely than previous datasets by using
3D environments.

**Strengths:**

The paper proposes a novel dataset, the EchoSet, which not only includes noise
and reverberation, but also object occlusion and different materials through
3D environments. The proposed speech separation architecture achieves high
accuracy on this new dataset while having few trainable parameters.

**Weaknesses:**

There are many issues with the experimental part of the paper, especially the
evaluation of the proposed model architecture.

The first issue is the datasets that were chosen. The main dataset for
speech separation, the WSJ0-2Mix, was not used. Instead, three others were - the proposed EchoSet, the LRS2-2Mix and the Libri2Mix. While choosing
the LRS2-2Mix over WHAMR! is motivated, the main problem concerns the
Libri2Mix. The results of the Libri2Mix reported here do no coincide with
previous reported results. The Conv-TasNet normally reaches 14.7 dB SI-SDRi
while this paper only report 12.1 dB SI-SDRi. Since all of the results reported
in this paper disagree with previous results, I assume that this is caused by
the choice of limiting the sequence length of the Libri2Mix to 3 seconds, as is
mentioned in the appendix. It also appears that this paper only uses utterances
from train-100 and leaves out utterances from train-360 as is used in the original
Libri2Mix. Therefore, this paper significantly changed the dataset of the Libri2Mix
while still calling it Libri2Mix. Since the training set was clearly significantly
altered, the results obtained here are very difficult to interpret because the
normal Libri2Mix contains more and significantly longer training data.

If one wanted to include two speaker speech separation, why not just use the
standard WSJ0-2Mix which basically everyone uses? And how are future papers
meant to compare themselves to the results reported in this paper? Is everyone
now meant to use this altered Libri2Mix? Since the source code is included,
others could run experiments to figure out the accuracy of the TIGER model
on the normal Libri2Mix or the WSJ0-2Mix. But it simply does not make sense
to have others go through said effort instead of having the original paper contain
this basic information.

There are, however, even more aspects of the results that are unusual. The
speed and memory measurements were taken using an input sequence containing
16k elements. This does not represent any of the datasets used. The EchoSet
is reported to use 6 second inputs, meaning it should contain 96k for 16 kHz,
the LRS2-2Mix uses 2 second inputs at 16 kHz, meaning a sequence length of
32k, and the Libri2Mix uses 3 second inputs, meaning a sequence length of 48k
at 16 kHz. Choosing any (or preferably all) of the sequence lengths used in the
datasets would be logical, but showing the results of a much shorter sequence
length is not.

However, even if all results reported in the paper were correct, they would
still not make a convincing case for the TIGER architecture. The biggest
strength of it is its low model size. While the experiments concerning accuracy
as well as computational cost are flawed as described above, the results of the
TIGER architecture are still not very convincing except for its accuracy on
the EchoSet. It is neither particularly fast nor memory efficient and was only
compared to a single other recent SOTA architecture, the TF-GridNet, which
it cannot match in accuracy in 2 of the 3 datasets tested.
While the paper includes an ablation study, it doesn’t answer the most
pressing questions. What kind of performance (in terms of accuracy/speed/memory
usage) would the TF-GridNet have if it shared model weights across its blocks
and reduced some parameters to match the TIGER’s model size? If such a
comparison was made and the TIGER architecture would be more accurate/faster/used
less memory, then there would be a better argument for it. As of now, it just
looks like it sacrifices accuracy for model size.

The paper also fails to mention perhaps the most significant previous work
for speech separation models with few trainable parameters, which is Group
communication [1]. It is incorrectly claimed that the TIGER architecture is
the first speech separation model with less than 1 million parameters, when the
aforementioned paper includes models with less than 100k parameters and is
roughly 4 years old.

The paper also claims that the models trained on the EchoSet perform better
on real world signals than the Libri2Mix and LRS2-2Mix. This might be true,
but is difficult to say with certainty due to the lack of information in the paper.
The real world data that was used is described in the appendix - however,
important details like its sampling rate and duration were left out. The sampling
rate for the speech separation datasets is only specified for the LRS2-2Mix to be
16 kHz but I assume it to be the same for the Libri2Mix and the EchoSet since
the training configuration appendix talks of frequency ranges from 0-8 kHz.
The duration for these datasets, however, is very different - 2 seconds for the
LRS2-2Mix, 3 seconds for the Libri2Mix and 6 seconds for the EchoSet. If the
average duration of the real world data is also closer to 6 seconds, then it would
give the EchoSet an unfair advantage over the other datasets. Again, this might
not be the case, but without clarification of the duration and sampling rate of all
the data used the results given cannot be verified. In fact, the reported results
in Figure 4 are very unexpected. How is the LRS2-2Mix often even worse than
the Libri2Mix (despite using background noise and reverb, same as the real
world data) while the EchoSet is massively better than both? The paper never
attempts to explain this oddity.

References

[1] Yi Luo, Cong Han, and Nima Mesgarani. “Group Communication With
Context Codec for Lightweight Source Separation”. In: IEEE/ACM Transactions
on Audio, Speech, and Language Processing 29 (2020), pp. 1752–1761.

**Questions:**

1. The assembly of the real world data is somewhat confusing. As I understand
it, the utterances as well as the noise was rerecorded in the real world and then
mixed together at different SDRs. Why did you not play multiple utterances
and noise from different locations in the room at the same time which would
resemble a real world mixture more closely rather than mixing them together
afterwards?
2. Computational cost was measured on a one second input at 16 kHz.
This choice is never motivated and in the context of speech separation illogical.
While there are some previous papers that have done the same [2, 3], neither
of them had any justification for this choice either. The standard for speech
separation would be an input of at least 4 seconds [4, 5]. The reasoning is
that the WSJ0-2Mix, contains utterances with an average length of about 4-5
seconds. The EchoSet, meaning the dataset the paper proposes, uses 6 second
utterances. Why choose a completely different length for the calculation of
speed and memory usage?
3. The sampling rate for the speech separation datasets is only specified
for the LRS2-2Mix as 16 kHz. Is it the same for the other two (Libri2Mix and
Echoset)? If so, why? Typically, 8 kHz is used and while there is nothing to be
said against using 16 kHz, it still makes future comparisons difficult to not also
include 8 kHz data.

References

[2] Kai Li, Runxuan Yang, and Xiaolin Hu. “An efficient encoder-decoder
architecture with top-down attention for speech separation”. In: ICLR.
2023.

[3] Chen Chen et al. “A Neural State-Space Modeling Approach to Efficient
Speech Separation”. In: Proc. INTERSPEECH 2023. 2023, pp. 3784–3788.
doi: 10.21437/Interspeech.2023-696.

[4] Zhong-Qiu Wang et al. TF-GridNet: Integrating Full- and Sub-Band Modeling
for Speech Separation. 2023. arXiv: 2211.12433 [cs.SD].

[5] Cem Subakan et al. Exploring Self-Attention Mechanisms for Speech Separation.
2023. arXiv: 2202.02884 [eess.AS].

---

> ### Author Response · Authors · 2024-11-22
>
> We appreciate the time and effort you have devoted to reviewing our work and offering your feedback. Below, we have addressed each of your points in detail.
>
> ***Q1 The first issue is the datasets that were chosen. The main dataset for speech separation, the WSJ0-2Mix, was not used. Why not just use the standard WSJ0-2Mix which basically everyone uses?***
>
> **A1** The WSJ0-2Mix dataset was indeed a widely used dataset in the speech separation research community. But due to its lack of noise and reverberation, it presents a relatively simplified separation task that fails to capture the complexity of real-world scenarios. While many models achieve excellent performance on WSJ0-2Mix, their separation performance often degrades significantly in real-world environments. For instance, [1] points out "even though Conv-TasNet's separation quality is close to perfect on wsj0-2mix, the ability to generalize to speech coming from a wider range of speakers and recorded in slightly different conditions has not yet been achieved". Similarly, the model in [2] demonstrates limitations in generalization when trained on WSJ0-2Mix, struggling to adapt to other datasets. Therefore, we need a dataset which more realistically reflects real-world acoustic effects in order to facilitate model performance on real-world audio.
>
> Moreover, recent recent studies [3-8] have used more complex datasets instead of WSJ0-2Mix to evaluate the model performance in a more realistic way. Based on these observations, we opted to use more diverse and challenging datasets to validate the effectiveness of our approach. This choice ensures a closer alignment with practical application scenarios.
>
> ***Q2 The reported result of Conv-TasNet on Libri2Mix do no coincide with previous reported results.  The training set of Libri2Mix was clearly significantly altered.***
>
> **A2** We would like to clarify that we used Libri2Mix train-100 min mode at 16 kHz in our experiments. We have added it in both Section 5 and Appendix A in the revised version. The Conv-TasNet performance (14.7 dB SI-SDRi) mentioned by the reviewer under the 8 kHz Libri2Mix train-360 min setting is different from our experimental setup.
>
> It is worth noting that 16 kHz audio sampling rate in Libri2Mix has been widely adopted in several speech separation studies [9-14], and TIGER is not the first to utilize it. We chose a 16 kHz sampling rate because it is more practical for real-world scenarios, as usually modern mobile devices have a minimum recording sampling rate of 16 kHz.
>
> Regarding the dataset size, we chose train-100 instead of train-360 due to considerations of training cost. Libri2Mix train-100 is an official standard subset (see [LibriMix GitHub repository](https://github.com/JorisCos/LibriMix)), making it a reasonable choice. In fact, there are other works that only used the Libri2Mix train-100 subset for training [8,15,16].
>
> To address the reviewer's concerns more comprehensively, we conducted additional experiments using the Libri2Mix train-360 min setting while maintaining the 16 kHz sampling rate. The table below presents the performance of TIGER (large), TF-GridNet, and BSRNN under this setting. On Libri2Mix train-360, TIGER achieved the performance about 1 dB lower than that of TF-GridNet on both SI-SDRi and SDRi, with only 5% parameters and  computational costs. The difference of performance is similar to the results when models are trained on Libri2Mix train-100 min, which further confirms that the results obtained using train-100 training set are reliable.
>
> | Model         | SI-SDRi (dB) | SDRi (dB) |
> |---------------|---------|------|
> | BSRNN         | 15.2    | 15.5 |
> | TF-GridNet    | 20.1    | 20.4 |
> | TIGER (large) | 19.0    | 19.4 |
>
> For audio length settings, we followed established conventions and set the training audio length to 3 seconds, as seen in configurations like the Asteroid project's default setup (see [Asteroid configuration](https://github.com/asteroid-team/asteroid/blob/master/egs/librimix/ConvTasNet/local/conf.yml)). Furthermore, we plan to release all datasets, relevant code, and configuration files to facilitate reproducibility.

---

> ### Author Response · Authors · 2024-11-22
>
> ***Q3 What kind of performance (in terms of accuracy/speed/memory usage) would the TF-GridNet have if it shared model weights across its blocks and reduced some parameters to match the TIGER’s model size?***
>
> **A3** We shared model weights across the blocks of TF-GridNet and reduced some parameters to match the TIGER’s model size. The models are trained on LRS2-2Mix. The window and hop size of STFT and iSTFT were set to 128 and 64. The number of TF-GridNet blocks was 3. The hidden size of LSTM was 132. The embedding dimension for each T-F unit was 48, and the kernel size for Unfold and Deconv1D is 4. Other parameters were set as the original paper. Here is the result. The effiency is evaluated on 1-second and 6-second audio during inference stage.
>
> | Model        | SI-SDRi (dB) | SDRi (dB)  | Params (M) | MACs (G)           | CPU Time (ms)     | GPU Time (ms)      | GPU Memory (MB)       |
> |--------------|---------|-------|--------|--------------------|-------------------|--------------------|-----------------------|
> | TF-GridNet (small)   | 12.9    | 13.3  | 0.82   | 39.09/258.20    | 1271.50/5473.09   | 86.14/271.90  | 376.24/2180.24    |
> | TIGER (large)  | 15.1    | 15.3  | 0.82   | 15.27/113.73      | 765.47/3876.61   | 74.51/158.05         | 122.23/640.13       |
>
> From the table, we can see that when TF-GridNet and TIGER have similar parameter size, TIGER performs better than TF-GridNet and is more efficient.
>
> ***Q4 The paper also fails to mention perhaps the most significant previous work for speech separation models with few trainable parameters, which is Group communication. It is incorrectly claimed that the TIGER architecture is the first speech separation model with less than 1 million parameters, when the aforementioned paper includes models with less than 100k parameters and is roughly 4 years old.***
>
> **A4** We acknowledge that the relevant statements in the paper do contain inaccuracies, and we have removed this sentence to avoid misrepresentation. However, it should be noted that the group communication (GC) approach is mainly used as a parameter compression technique in existing model architectures, whereas our approach focuses on improving it at the level of network architecture design. Compared to the current SOTA model TF-GridNet, TIGER proposed by us is competitive in terms of separation performance, while its parameter amount is reduced by 94.3% and MACs (multiply-accumulate operations) by 95.3%.
>
> In addition, to further explore the lightweighting potential of the model, we supplement the experimental results of a smaller version of the TIGER model (TIGER Tiny) as well as the compressed SudoRM-RF model based on the GC3 method on the EchoSet dataset (as shown in the table below.) The hyperparameters of the TIGER Tiny are set as follows: the input channels are reduced from 128 to 24, the output channels are reduced from 256 to 64, and the number of FFI blocks is kept unchanged; the hyperparameter settings for SudoRM-RF are strictly in accordance with the publicly available configurations of GC3 (see https://github.com/yluo42/GC3). The following table details the experimental results. The efficiency is evaluated on 1s and 6s audio input during inference stage.
>
> | Method           | SI-SDRi (dB) | SDRi (dB)  | MACs (G)    | Params (K) | CPU Time (ms)     | GPU Time (ms)     | GPU Memory (MB)     |
> |------------------|---------|-------|-------------|------------|-------------------|-------------------|---------------------|
> | SudoRM-RF GC3    | 4.6     | 5.0   | 0.81/4.84   | 303.57     | 123.67/612.38     | 32.68/58.35       | 25.20/151.25        |
> | TIGER Tiny       | 10.4    | 10.7  | 0.72/7.48   | 102.12     | 219.89/867.61     | 30.21/55.70       | 25.44/153.12        |
>
> As shown in the table, TIGER Tiny significantly outperforms SudoRM-RF GC3 at comparable efficiencies, which proves that TIGER is a very effective lightweight model.

---

> ### Author Response · Authors · 2024-11-22
>
> ***Q5 The real world data that was used is described in the appendix - however, important details like its sampling rate and duration were left out.***
>
> **A5** This was indeed our oversight. We have added the related details to Appendix A in  the paper. The Real-world dataset is a small evaluation dataset collected from real-world scenarios, and its data collection process is described as follows. First, we selected 10 rooms of varying sizes and shapes as distinct test environments. Then, we randomly sampled approximately 1.5 hours of 16kHz speech audio from the LibriSpeech test set (the length of each audio is 60s), with noise data sourced from the WHAM! noise dataset. During the recording process, audio content was played using the speakers of a 2023 MacBook Pro and recorded via a Logitech Blue Yeti Nano omnidirectional microphone placed in a fixed position. The distance between speakers and the microphone is not fixed, ranging from 0.3m to 2m. The recording parameters were set to a 16kHz sampling rate and 32-bit depth. This setup ensured that both speech and noise were recorded in the same room, preserving the authenticity of the reverberation effects. Finally, we processed the collected audio by mixing the recordings. Specifically, audio from different speakers was mixed using signal-to-noise ratios (SNRs) randomly sampled between -5 dB and 5 dB. Noise data was added using SNRs randomly sampled between -10 dB and 10 dB. Since the propagation paths of sounds in the air are independent of one another, mixing these components is considered a reasonable approach. This design ensures the realism and diversity of the evaluation dataset, effectively capturing the complexity of speech separation in real-world conditions.
>
> ***Q6 The duration for these datasets, however, is very different - 2 seconds for the LRS2-2Mix, 3 seconds for the Libri2Mix and 6 seconds for the EchoSet. If the average duration of the real world data is also closer to 6 seconds, then it would give the EchoSet an unfair advantage over the other datasets.***
>
> **A6** In fact, we trained on the EchoSet dataset by randomly sampling a 3-second segment from the 6-second-long audio, rather than using full-length (6-second) audio. This setup is described in Appendix B in the paper. The training length of the EchoSet dataset is aligned with the length of Libri2Mix, which ensures fairness in its comparison.
>
> In addition, we recorded real-world audio of 60 seconds in length. In the inference phase, we used the training lengths of the respective datasets to inference the 60-second audio with a 50% overlap sliding scale. This approach to some extent mitigates the problem of model performance degradation that may be caused by the difference in training and inference lengths, thus ensuring fairness in the model's performance in the inference phase.
>
> ***Q7 How is the LRS2-2Mix often even worse than the Libri2Mix (despite using background noise and reverb, same as the real world data) while the EchoSet is massively better than both? The paper never attempts to explain this oddity.***
>
> **A7** As we mentioned in Section 2 of the paper, the LRS2-2Mix dataset was generated by mixing video clips provided by the BBC. Since these video clips were recorded in different environments, mixing their audio directly would result in an inconsistent acoustic environment, and therefore would not generate audio with realistic reverberation effects. This unrealistic reverberation explains why models trained on the LRS2-2Mix dataset often perform poorly on real datasets.
>
> In contrast, the EchoSet dataset, when constructing audio with reverb and noise, ensures that all mix components are in the same acoustic environment, thus more closely resembling the real scene. This design allows models trained on EchoSet to show better generalization on real-world data.
>
> ***Q8 The assembly of the real world data is somewhat confusing. As I understand it, the utterances as well as the noise was rerecorded in the real world and then mixed together at different SDRs. Why did you not play multiple utterances and noise from different locations in the room at the same time which would resemble a real world mixture more closely rather than mixing them together afterwards?***
>
> **A8** The reason we used this recording method is that real labels need to be obtained in order to compute SI-SDRi and SDRi, which are important metrics used for model performance comparisons. These metrics cannot be accurately obtained if the mixed audio is recorded directly. Therefore, we generate mixed audio by combining separate recordings. It is reasonable and effective, because sound propagates independently through the air in accordance with the principle of wave superposition [17]. Generating mixed audio in this way preserves the labeling information of the individual sound sources, thus providing a reliable benchmark for model evaluation.

---

> ### Author Response · Authors · 2024-11-22
>
> ***Q9 Computational cost was measured on a one second input at 16 kHz. This choice is never motivated and in the context of speech separation illogical. While there are some previous papers that have done the same [2, 3], neither of them had any justification for this choice either. The standard for speech separation would be an input of at least 4 seconds [4, 5]. The reasoning is that the WSJ0-2Mix, contains utterances with an average length of about 4-5 seconds. The EchoSet, meaning the dataset the paper proposes, uses 6 second utterances. Why choose a completely different length for the calculation of speed and memory usage?***
>
> **A9** It is important to note from the outset that there is no standardized practice for the evaluation of model efficiency. As mentioned by the reviewer, some studies use 1 second of audio for testing, while others use 4 seconds of audio. However, the efficiency of the model is independent of the audio length of the dataset used. In our study, all models were consistently tested using 1-second length audio, so this comparison is fair and reliable.
>
> We have supplemented our evaluation results for 6-second audio for reference. The experimental results are shown in the following. As shown in the table, regardless of the duration of audio, TIGER has lower latency and memory usage than models with similar performance.
>
> In the table, the audio length for evaluating efficiency is 1/6 s. 'OOM' stands for out of memory (>24564MB).
> | Model       | MACs(G)            | Training GPU Time (ms)   | Training GPU Memory (MB)   | Inference CPU Time (ms)   | Inference GPU Time (ms)  | Inference GPU Memory (MB)   |
> | ----------- | -------------------- | ------------------------ | -------------------------- | ------------------------- | ------------------------- | ---------------------------- |
> | Conv-TasNet | 7.19/43.16           | 92.96/137.32             | 1436.94/8556.42            | 64.21/1325.22             | 13.17/26.53              | 28.78/170.47                |
> | DualPathRNN | 45.01/259.07         | 67.23/296.90             | 1813.55/10242.49           | 723.13/2780.52            | 30.38/157.05             | 298.09/2095.14              |
> | SudoRM-RF   | 4.65/27.91           | 118.46/151.45            | 1353.43/8040.96            | 104.32/571.84             | 20.66/36.15              | 24.42/146.88                |
> | A-FRCNN     | 81.28/487.60         | 230.53/366.78            | 2925.83/16708.81           | 478.58/1840.77            | 82.65/101.19             | 163.82/433.00               |
> | TDANet      | 9.19/54.85           | 263.43/OOM               | 4260.36/OOM                | 434.44/1819.83            | 74.27/120.93             | 136.96/287.84               |
> | BSRNN       | 98.70/587.34         | 258.55/519.65            | 1093.11/10297.47           | 897.27/3821.08            | 78.27/170.56             | 130.24/771.24               |
> | TF-GridNet  | 323.75/2004.02       | 284.17/OOM               | 5432.94/OOM                | 2019.60/11291.58          | 94.30/498.45             | 491.73/4771.31              |
> | TIGER Small | 7.65/56.94           | 160.17/402.09            | 2049.46/11503.89           | 351.15/2226.12            | 42.38/110.29             | 122.23/635.96               |
> | TIGER Large | 15.27/113.73         | 229.23/544.26            | 3989.59/22831.03           | 765.47/3876.61            | 74.51/158.05             | 122.23/640.13               |
>
> ***Q10 The sampling rate for the speech separation datasets is only specified for the LRS2-2Mix as 16 kHz. Is it the same for the other two (Libri2Mix and Echoset)? If so, why? Typically, 8 kHz is used and while there is nothing to be said against using 16 kHz, it still makes future comparisons difficult to not also include 8 kHz data.***
>
> **A10** The sampling rate for all the three datasets (Libri2Mix, EchoSet and LRS2-2Mix) is 16 kHz, which we have clarified in both Section 5 and Appendix A in the paper. As answered in **Q2**, 16 kHz audio sampling rate in Libri2Mix has been widely adopted in several speech separation studies [9-14], and TIGER is not the first to utilize it. Besides, modern mobile device microphones usually record at a minimum sampling rate of 16 kHz nowadays. High sampling rate data is the future trend in speech research, so we used 16 kHz data rather than 8 kHz.

---

> ### Author Response · Authors · 2024-11-22
>
> ***References***
>
> [1] Cosentino J, Pariente M, Cornell S, et al. Librimix: An open-source dataset for generalizable speech separation. arXiv preprint arXiv:2005.11262, 2020.
>
> [2] Kadıoğlu B, Horgan M, Liu X, et al. An empirical study of Conv-TasNet. ICASSP 2020-2020 IEEE International Conference on Acoustics, Speech and Signal Processing (ICASSP). IEEE, 2020: 7264-7268.
>
> [3] Wang W, Pan Z, Li X, et al. Speech separation with pretrained frontend to minimize domain mismatch. IEEE/ACM Transactions on Audio, Speech, and Language Processing, 2024.
>
> [4] Liu D, Zhang T, Christensen M G, et al. Multi-layer encoder–decoder time-domain single channel speech separation. Pattern Recognition Letters, 2024, 181: 86-91.
>
> [5] Li K, Yang R, Hu X. An efficient encoder-decoder architecture with top-down attention for speech separation. The Eleventh International Conference on Learning Representations.
>
> [6] Pons J, Liu X, Pascual S, et al. Gass: Generalizing audio source separation with large-scale data. ICASSP 2024-2024 IEEE International Conference on Acoustics, Speech and Signal Processing (ICASSP). IEEE, 2024: 546-550.
>
> [7] Aung A N, Liao C W, Hung J W. Effective Monoaural Speech Separation through Convolutional Top-Down Multi-View Network. Future Internet, 2024, 16(5): 151.
>
> [8] Aung A N, Hung J. Improving Top-Down Attention Network in Speech Separation by Employing Hand-Crafted Filterbank and Parameter-Sharing Transformer. Electronics, 2024, 13(21): 4174.
>
> [9] Li Y. Effectiveness of SSL Representations for Source Separation. PhD thesis.
>
> [10] Berger E, Schuppler B, Pernkopf F, et al. Single Channel Source Separation in the Wild–Conversational Speech in Realistic Environments. Speech Communication; 15th ITG Conference. VDE, 2023: 96-100.
>
> [11] Xu A, Choudhury R R. Learning to separate voices by spatial regions. International Conference on Machine Learning. PMLR, 2022: 24539-24549.
>
> [12] Wang K, Yang Y, Huang H, et al. Speakeraugment: Data augmentation for generalizable source separation via speaker parameter manipulation. ICASSP 2023-2023 IEEE International Conference on Acoustics, Speech and Signal Processing (ICASSP). IEEE, 2023: 1-5.
>
> [13] Erdogan H, Wisdom S, Chang X, et al. Tokensplit: Using discrete speech representations for direct, refined, and transcript-conditioned speech separation and recognition. arXiv preprint arXiv:2308.10415, 2023.
>
> [14] Wisdom S, Tzinis E, Erdogan H, et al. Unsupervised speech separation using mixtures of mixtures. ICML 2020 Workshop on Self-supervision in Audio and Speech. 2020.
>
> [15] Fazel-Zarandi M, Hsu W N. Cocktail Hubert: Generalized Self-Supervised Pre-Training for Mixture and Single-Source Speech. ICASSP 2023-2023 IEEE International Conference on Acoustics, Speech and Signal Processing (ICASSP). IEEE, 2023: 1-5.
>
> [16] Zhang Z, Chen C, Chen H H, et al. Noise-Aware Speech Separation with Contrastive Learning. ICASSP 2024-2024 IEEE International Conference on Acoustics, Speech and Signal Processing (ICASSP). IEEE, 2024: 1381-1385.
>
> [17] Kleiner, Mendel. (2011). Acoustics and Audio Technology.

---

> ### Author Response · Authors · 2024-11-29
> **Seeking Your Thoughts on Our Response**
>
> Dear Reviewer,
>
> Thank you for your detailed feedback on our work. We have made revisions to address your comments and shared our responses in the discussion. We are keen to know if our updates meet your expectations and would welcome any further discussion. We deeply appreciate your time and effort.
>
> Authors

---

### Official Review · Reviewer_Sqdv · 2024-11-05

**Soundness:** 3
**Presentation:** 3
**Contribution:** 3
**Rating:** 6
**Confidence:** 4

**Summary:**

In this paper, the authors propose a speech separation setup which relies on multi-scale selective attention and interleaved time-frequency attention modules. The design is relatively lightweight, significantly reducing the number of model parameters and consequently, computing resources in inference. They also introduced   a speech separation corpus, EchoSet, that allegedly has more realistic noise and reverberation. They conducted comprehensive empirical evaluations of several comparable models and public data sets.

**Strengths:**

The paper is well-written overall, following a logical organization and making sound arguments. The development of the proposed system is easy to follow. The details of the modules inside the systems are adequately presented. The evaluation results, given the nature of the paper as an empirical study, can seem a bit under-developed, but the major evaluations are made and presented. The ablation studies are important to help readers understand the importance of the MFA and F3A modules. That being said, similar ablation studies can also be performed on the EchoSet, to determine which aspect of the simulation set this new corpus apart from the former ones.
The paper highlighted the strong performance of the proposed speech separation system and its small footprint in terms of model size.

**Weaknesses:**

The comparison between the various corpus is not strictly fair, given the data sets have different sizes. From the data provided by the authors, EchoSet contains 33.78 hrs, whereas LRS2-2Mix has 11.1 hours and Libri2Mix has 58 hrs. Did the authors augment/select the data to ensure that the same amount of data is used in training to evaluate the systems in Table 2?
The argument in Sect. 6.1 is not entirely clear to me. What does data collected in the real world refer to? Is there any description of this data?
To measure the model complexity, model parameters are acceptable, but MACs is inadequate. To go beyond theoretical analysis, the authors can measure the memory usage, power consumption, throughput and latency on server or edge devices.
Related to the comments above, Sect. 6 can be beefed up with more analysis. For example, does TIGER or models trained with EchoSet perform better in lower SNR/stronger reverb compared to other models? Can we break it down? How does Tiger/EchoSet fare on male-male pair vs. male-female pair, etc.?

**Questions:**

In Table 4, how does LowFreqNarrowSplit compare with Mel-split?
How scalable is Tiger for multi-talker separation?
How scalable is Tiger for audio separation beyond speech, as alluded to in Table 9, where music and environmental sound is still a lot harder than speech. What's the path forward?

---

> ### Author Response · Authors · 2024-11-22
>
> We sincerely appreciate the time you've taken to review our work. Below, we have responded to each of your comments in detail.
>
> ***Q1 The comparison between the various corpus is not strictly fair, given the data sets have different sizes. From the data provided by the authors, EchoSet contains 33.78 hrs, whereas LRS2-2Mix has 11.1 hours and Libri2Mix has 58 hrs. Did the authors augment/select the data to ensure that the same amount of data is used in training to evaluate the systems in Table 2?***
>
> **A1** The training set hours on EchoSet is about half that of Libri2Mix. However, models trained on EchoSet perform distinctly better than models trained on Libri2Mix, which shows that our data is closer to the real audio than Libri2Mix.
>
> As for the fair comparison between LRS2-2Mix and EchoSet, we randomly selected one-third of the data volume from EchoSet training set, containing 11.26 hours of data. We trained TIGER (large), BSRNN and TF-GridNet, and evaluated it on the EchoSet test set and real-world data. The following are the results. Even if only 1/3 of the EchoSet training data is used, the models' performance on real-world data is still significantly better than training on LRS2-2Mix, which proves that EchoSet is indeed closer to reality and more effective.
>
> Training on 1/3 EchoSet:
> | Model         | EchoSet SI-SDRi (dB) | EchoSet SDRi (dB) | Real-world data SI-SDRi (dB) | Real-world data SDRi (dB) |
> |---------------|-----------------|--------------|-------------------------|----------------------|
> | TIGER (large) | 13.10           | 13.64        | 4.97                    | 5.70                 |
> | TF-GridNet    | 12.31           | 13.22        | 3.75                    | 4.41                 |
>
> Training on LRS2-2Mix:
> | Model         | LRS2-2Mix SI-SDRi (dB) | LRS2-2Mix SDRi (dB) | Real-world data SI-SDRi (dB) | Real-world data SDRi (dB) |
> |---------------|-------------------|----------------|-------------------------|----------------------|
> | TIGER (large) | 15.1              | 15.3           | 1.49                    | 2.77                 |
> | TF-GridNet    | 15.4              | 15.7           | 0.29                    | 1.85                 |
>
>
> ***Q2 The argument in Sect. 6.1 is not entirely clear to me. What does data collected in the real world refer to? Is there any description of this data?***
>
> **A2** The real-world dataset is a small evaluation dataset collected from real-world scenarios, and its data collection process is described as follows. First, we selected 10 rooms of varying sizes and shapes as distinct test environments. Then, we randomly sampled approximately 1.5 hours of 16kHz speech audio from the LibriSpeech test set, with noise data sourced from the WHAM! noise dataset. During the recording process, audio content was played using the speakers of a 2023 MacBook Pro and recorded via a Logitech Blue Yeti Nano omnidirectional microphone placed in a fixed position. The distance between the speaker and the microphone was randomly selected from 0.3m to 2m. The recording parameters were set to a 16kHz sampling rate and 32-bit depth. This setup ensured that both speech and noise were recorded in the same room, preserving the authenticity of the reverberation effects. Finally, we processed the collected audio by mixing the recordings. Specifically, audio from different speakers was mixed using signal-to-distortion ratios (SDRs) randomly sampled between -5 dB and 5 dB. Noise data was added using SDRs randomly sampled between -10 dB and 10 dB. Since the propagation paths of sounds in the air are independent of one another, mixing these components is considered a reasonable approach. This design ensures the realism and diversity of the evaluation dataset, effectively capturing the complexity of speech separation in real-world conditions. We have supplemented the collection details of the real-world dataset in Appendix A.
>
> ***Q3 To measure the model complexity, model parameters are acceptable, but MACs is inadequate. To go beyond theoretical analysis, the authors can measure the memory usage, power consumption, throughput and latency on server or edge devices.***
>
> **A3** We have measured the memory usage and latency on the server. Please refer to Table 3 in the paper. We reported the GPU time and GPU memory usage during the training process, as well as the CPU time, GPU time, and GPU memory usage during the inference process. To simulate the limited computational conditions of mobile devices on which the speech separation model is deployed in real-world situations, we fixed the number of threads to 1 when calculating CPU (Intel(R) Xeon(R) Gold 6326) time and only used a single card when calculating GPU (GeForce RTX 4090) time.

---

> ### Author Response · Authors · 2024-11-22
>
> ***Q4 Does TIGER trained with EchoSet perform better in lower SNR/stronger reverb compared to other models? Can we break it down?***
>
> **A4** To quickly evaluate the impact of different signal-to-noise ratios (SNR) on the model's performance, we randomly selected 300 audio samples from the EchoSet test set and mixed them with noise at various SNR levels (-10, -5, 0, 5 dB). We tested TIGER (large) along with the two best baseline models (BSRNN and TF-GridNet). The results are shown in the table below.
>
> | Method      | SI-SNRi (dB)                       | SNRi (dB)                        |
> |-------------|------------------------------|------------------------------|
> | BSRNN       | -4.19/-0.18/2.27/5.51        | 2.47/4.69/6.65/10.76         |
> | TF-GridNet  | -3.16/0.52/4.16/6.05         | 2.65/4.96/7.25/10.91         |
> | TIGER  (large)     | -3.05/1.25/5.49/7.19         | 3.20/6.13/8.93/11.19         |
>
> The results indicate that all models perform poorly at an SNR of -10 dB. However, under all noise conditions tested, TIGER consistently achieves the best separation results compared to BSRNN and TF-GridNet.
>
> Regarding conditions with strong reverberation, as EchoSet's reverberation is generated using 3D scene-based simulations and cannot be easily controlled, we did not provide related results. However, the audio data recorded in real-world environments includes room scenarios of varying sizes, thereby reflecting diverse reverberation conditions. As illustrated in Figure 4, models trained on the EchoSet dataset exhibit superior generalization capabilities compared to those trained on other datasets. And among all the models, TIGER achieved the best performance. This result highlights the advantages of the EchoSet dataset, demonstrating that its design better encompasses the diverse acoustic conditions encountered in practical applications, as well as the superior generalization ability of TIGER to complex acoustic conditions compared with other models.
>
> ***Q5 How does Tiger/EchoSet fare on male-male pair vs. male-female pair, etc.?***
>
> **A5** We divided the EchoSet test set into male-male, female-female, and male-female categories, and tested the performance using TIGER (large) respectively. The following table shows the result.
>
> | Gender of speakers | Count | SI-SDRi (dB) | SDRi (dB) |
> |---------------------|-------|--------------|-----------|
> | male-male           | 637   | 12.76        | 13.30     |
> | female-female       | 644   | 14.29        | 14.71     |
> | male-female         | 1369  | 13.93        | 14.41     |
>
> According to the table, TIGER has similar performance in audio separation for male-male, male-female, and female-female pairings. Male voices are a bit more difficult to separate than female voices.
>
> ***Q6 In Table 4, how does LowFreqNarrowSplit compare with Mel-split?***
>
> **A6** The Mel-split approach, as we understand it, achieves frequency compression by remapping frequencies onto the Mel scale, enabling audio separation tasks. However, few studies have explored performing audio separation directly on the Mel scale. The primary reason for this is that the Mel spectrogram is a nonlinear transformation, making it challenging to map back to the original spectral features and thereby increasing the learning difficulty for models. In contrast, the band-splitting method we adopt is based on linear compression, which facilitates reconstructing band features back into the original spectral features with greater ease.
>
> ***Q7 How scalable is Tiger for multi-talker separation?***
>
> **A7** To demonstrate the performance on a larger number of speakers, we conducted experiments on the train-100 subset of the LibriMix 16kHz dataset for scenarios involving three speakers. We trained and evaluated TF-GridNet, BSRNN and TIGER (large). The training and evaluation configurations kept the same as described in Appendix B and C in the paper. The table shows their performance. The results indicate that as the number of speakers increases, the performance of all models experiences a certain degree of degradation. Under the setting of 3 speakers, TIGER still exhibits performance slightly lower to TF-GridNet, with significantly reduced parameters and computational costs.
>
> | Method     | SDRi  | SI-SDRi |
> |------------|-------|---------|
> | BSRNN      | 9.43  | 9.77    |
> | TF-GridNet | 12.30 | 12.52   |
> | TIGER      | 11.35 | 11.64   |

---

> ### Author Response · Authors · 2024-11-22
>
> ***Q8 How scalable is Tiger for audio separation beyond speech, as alluded to in Table 9, where music and environmental sound is still a lot harder than speech. What's the path forward?***
>
> **A8** We consider that the TIGER model demonstrates superior scalability. Even TIGER (large) maintains significantly lower parameter counts and computational complexity compared to existing speech separation models. Furthermore, TIGER has shown outstanding performance in three-track sound separation tasks, significantly outperforming BSRNN and other models. Expanding the model's parameter scale is expected to further enhance separation performance.
>
> To make the model more effective for real-world applications, we suggest the following three directions for future improvements:
>
> 1. Development of Realistic Separation Data. Enhancing the realism of datasets is crucial for improving model performance. This can be achieved through the following approaches. Audio Data Preprocessing: For instance, reconstructing low-sample-rate audio into high-sample-rate audio to enhance quality and detail. Expanding Music Data: Leveraging music separation models to isolate vocals from existing music data, thus avoiding vocal interference and enriching the training data for source separation. Voice Simulation Tools: Developing tools capable of simulating reverberation effects in real-world scenarios to generate data that closely mirrors authentic environments.
>
> 2. Designing Efficient and High-Performance Separation Models. Efficient models can process more data within a fixed timeframe, thereby significantly enhancing model usability in real-world environments. Additionally, such models can substantially reduce training and deployment costs, promoting their practical application in real-world scenarios.
>
> 3. Adopting Semi-Supervised Training Approaches. Semi-supervised learning methods can effectively leverage unlabeled data by combining high-quality labeled data with model-generated unlabeled separation data. This approach expands the training data volume, enabling the model to better adapt to complex real-world applications and improve its generalization capabilities.

---

> ### Author Response · Authors · 2024-11-29
> **Request for Further Insights on Revised Submission**
>
> Dear Reviewer,
>
> Thank you for your thoughtful and constructive comments on our submission. We have worked diligently to revise the paper and have responded to each of your points. We hope that our efforts have addressed your concerns and would be grateful for the opportunity to discuss further. Thank you again for your time and support.
>
> Authors

---

### Official Review · Reviewer_MkhV · 2024-11-05

**Soundness:** 4
**Presentation:** 4
**Contribution:** 4
**Rating:** 8
**Confidence:** 3

**Summary:**

This paper presents TIGER, an efficient speech separation model with significantly reduced parameters and computational costs, optimized for low-latency applications. Leveraging frequency band segmentation and attention mechanisms, TIGER captures rich contextual information. The authors introduce EchoSet, a realistic dataset for evaluating model robustness in complex environments. TIGER achieves state-of-the-art performance with 94.3% fewer parameters and 95.3% fewer MACs than existing models.

The result is very impressive and the details is rich. They also provide a high quality dataset which would be beneficial to the field.

**Strengths:**

1. Provide a new way of speech separation with very efficient paramater usage. It is very important to the industry and would be very influential
2. provide a new dataset which provides different content compared to all other similar kinds before.

**Weaknesses:**

1. The model focused on two speaker separation which makes the task much simpler. May need to provide a way to generalize the framework.

**Questions:**

1. How would the model deal with the audio with background noise?
2. Does the method still work when two people speaking in different setup? e.g. one people is much further to the speaker than another.
3. How should the model accommodate variable number of speakers?

---

> ### Author Response · Authors · 2024-11-22
>
> Your thoughtful review and constructive suggestions are deeply appreciated. We have carefully considered each point and provided detailed responses below.
>
> ***Q1 How would the model deal with the audio with background noise?***
>
> **A1** Our method is based on supervised learning. The ground truth for both speakers in the three datasets we used doesn't include noise. This setup encourages the model to learn to separate the voices of the target speakers while filtering out background noise, as it is trained to reconstruct the clean speech signals.
>
> ***Q2 Does the method still work when two people speaking in different setup? e.g. one people is much further to the speaker than another.***
>
> **A2** During the generation of the EchoSet dataset, we considered scenarios where the two speakers are at different distances from the microphone. Specifically, the distance between each speaker and the microphone was randomly sampled between 1 m and 5 m. Therefore, the dataset inherently includes cases where one speaker is farther away while the other is closer. In the experimental results presented in Table 2 in the paper, we demonstrated that TIGER exhibited superior separation performance compared to other models under these more complex conditions.
>
> In real-world dataset (described in Appendix A in the paper), which is used to evaluate models' performance on data from the real world, the situation where one speaker is farther than the other is also included. The distance between the speaker and the microphone was randomly selected from 0.3 m to 2 m. This ensures the authenticity of real-world dataset and also makes the separation more challenging. On real-world dataset, TIGER also outperforms other models.
>
> ***Q3 How should the model accommodate variable number of speakers?***
>
> **A3** We would like to clarify that our contribution lies in proposing a lightweight speech separation architecture, which is not directly tied to handling a variable number of speakers.
> To demonstrate the performance on a larger number of speakers, we conducted experiments on the train-100 subset of the Libri2Mix 16kHz dataset for scenarios involving three speakers, training TF-GridNet, BSRNN, and TIGER (large). The training and evaluation configurations kept the same as described in Appendix B and C in the paper. The experimental results are presented in the table below. The results indicate that as the number of speakers increases, the performance of all models experiences a certain degree of degradation. Under the setting of 3 speakers, TIGER still exhibits performance slightly lower to TF-GridNet, with significantly reduced parameters and computational costs.
>
> | Method     | SDRi  | SI-SDRi |
> |------------|-------|---------|
> | BSRNN      | 9.43  | 9.77    |
> | TF-GridNet | 12.30 | 12.52   |
> | TIGER      | 11.35 | 11.64   |
>
> For scenarios involving a variable number of speakers, one potential solution is to adopt the MixIT training paradigm [1], which can be applied to different model architectures to better adapt them to such conditions.
>
> ***References***
>
> [1] Wisdom S, Tzinis E, Erdogan H, et al. Unsupervised sound separation using mixture invariant training. Advances in neural information processing systems, 2020, 33: 3846-3857.

---

> ### Author Response · Authors · 2024-11-29
> **Looking Forward to Further Discussion on Our Response**
>
> Dear Reviewer,
>
> We greatly appreciate your constructive review and the insightful comments you provided. In response, we have submitted detailed explanations addressing your concerns. We would be delighted to know if these responses resolve the issues you raised and look forward to any further discussion. Thank you once again for your valuable feedback.
>
> Authors

---

### Official Review · Reviewer_kFvq · 2024-11-05

**Soundness:** 3
**Presentation:** 3
**Contribution:** 2
**Rating:** 6
**Confidence:** 2

**Summary:**

This paper presents an approach named TIGER for speech separation, emphasizing both efficiency and performance. The proposed approach utilizes frequency band-splitting, selective attention modules, and F3A to manage both temporal and frequency information. In addition, the authors introduced a new data set named EchoSet, that considers the realistic environmental factors such as noise, reverberation, and varied speaker overlap ratios. Experimental evaluations reveal that the proposed approach achieves competitive results across multiple benchmarks.

**Strengths:**

1) The authors' provision of open-source code enables researchers in the field to reproduce and build upon this work.
2) The authors have a detailed ablation study, as it clarifies the impact and effectiveness of each proposed module.

**Weaknesses:**

1) The proposed approach does not demonstrate clear performance advantages over the current SOTA method, with a noticeable performance gap.
2) Although the authors claimed their approach is more lightweight, the comparison is not entirely fair. A comparison with other systems that specifically employ lightweight methods would provide a more accurate assessment of the model's efficiency.

**Questions:**

1) The authors mentioned that the band-split module is inspired by prior knowledge of music separation. it would be helpful to clarify if the targeted speech separation task has similar prior knowledge and to highlight any key differences or similarities between the band-split module used here and that in the original paper.
2) In Figure 2, the authors mention that "Residual connections are used to retain original features and reduce learning difficulty." Could you please indicate where these residual connections are depicted in the figure?
3) In Figure 4, the authors use a line chart to present the results. This may be misleading because the results on the three datasets are independent of each other. A different visualization style, such as a bar chart, might provide a clearer comparison.
4) What are the differences between TIGER (small) and TIGER (large), why do they have the same model size?
5) In the ablation study, it looks like the proposed modules appear to involve trade-offs among various performance criteria. Could the authors discuss how to balance this tradeoff and what could be the best to consider?

---

> ### Author Response · Authors · 2024-11-22
>
> We are grateful for the time and effort you have invested in evaluating our work and offering such valuable feedback. Below, we have addressed all of your concerns in detail.
>
> ***Q1 The proposed approach does not demonstrate clear performance advantages over the current SOTA method, with a noticeable performance gap.***
>
> **A1** Admittedly, TIGER (large) demonstrates approximately a 6% and 2% performance gap compared to the current SOTA model, TF-GridNet, on Libri2Mix and LRS2-2Mix respectively. But the motivation of our work is to design a lightweight model which is more applicable in computationally constrained real-world scenarios while maintaining high performance. TIGER significantly reduces the number of parameters by 94.3% and the MACs by 95.3% compared with TF-GridNet, and according to Table 3, for inference of one-second audio, TIGER (large) took about 1/3 of the CPU Time and 3/4 of the GPU Time compared with TF-GridNet, demonstrating a significant calculation compression effect. Besides, TIGER took up less memory during training and inference, making TIGER more suitable for deployment on devices with limited computational resources. These evidence shows that TIGER is indeed very lightweight compared with TF-GridNet.
>
> Besides, on EchoSet, TIGER (large) achieves a 3.5% and 6.8% improvement in SDRi and SI-SDRi, while using only 5% of TF-GridNet's computational cost. Based on the results in Figure 4,  models trained on EchoSet generalize better to real-world data than those trained on Libri2Mix or LRS2-2Mix, suggesting that EchoSet is the most acoustically realistic dataset among the three. Therefore, performance on EchoSet provides a better indication of a model's separation ability in practical scenarios.
>
> Additionally, according to Figure 4, regardless of the training dataset, TIGER consistently achieves the best separation performance on real-world data, demonstrating that its architecture offers superior generalization to real-world conditions compared to other models.
>
> ***Q2 Although the authors claimed their approach is more lightweight, the comparison is not entirely fair. A comparison with other systems that specifically employ lightweight methods would provide a more accurate assessment of the model's efficiency.***
>
> **A2** To compare with the system that specifically employs lightweight methods, we supplement the experimental results of a smaller version of the TIGER model (TIGER Tiny) as well as the compressed SudoRM-RF model based on the GC3 method [1] on the EchoSet dataset. The hyperparameters of the TIGER Tiny are set as follows: the input channels are reduced from 128 to 24, the output channels are reduced from 256 to 64, and the number of FFI blocks kept unchanged; the hyperparameter settings for SudoRM-RF are strictly in accordance with the publicly available configurations of GC3 (see https://github.com/yluo42/GC3). The following table details the experimental results. The efficiency is evaluated on 1-second and 6-second audio input during inference stage.
>
> | Method           | SI-SDRi (dB) | SDRi (dB)  | MACs (G)    | Params (K) | CPU Time (ms)     | GPU Time (ms)     | GPU Memory (MB)     |
> |------------------|---------|-------|-------------|------------|-------------------|-------------------|---------------------|
> | SudoRM-RF GC3    | 4.6     | 5.0   | 0.81/4.84   | 303.57     | 123.67/612.38     | 32.68/58.35       | 25.20/151.25        |
> | TIGER Tiny       | 10.4    | 10.7  | 0.72/7.48   | 102.12     | 219.89/867.61     | 30.21/55.70       | 25.44/153.12        |
>
> As shown in the table, TIGER Tiny significantly outperforms SudoRM-RF GC3 at comparable efficiencies, which proves that TIGER is a very effective lightweight model.

---

> ### Author Response · Authors · 2024-11-22
>
> ***Q3 The authors mentioned that the band-split module is inspired by prior knowledge of music separation. it would be helpful to clarify if the targeted speech separation task has similar prior knowledge and to highlight any key differences or similarities between the band-split module used here and that in the original paper.***
>
> **A3** Since both music and speech are audio signals, both of them can be converted into time-frequency domain representation. Therefore, on both tasks, band-split module can be used to significantly reduce computational complexity through the compression of time-frequency domain features (the frequency dimension), enabling a lightweight model design.
>
> It is important to highlight that TIGER and BSRNN exhibit notable differences in terms of the number of subbands and the bandwidth partitioning strategy, with the most critical distinction observed in the segmentation approach for the 0–1 kHz frequency range. In BSRNN, the frequency range below 1 kHz is divided into subbands with a bandwidth of 100 Hz [2]. By contrast, in TIGER, leveraging the typical frequency range of human speech, which spans from 85 Hz to 1100 Hz [3], each frequency sample is treated as an independent subband. Consequently, the frequency range below 1 kHz is divided into subbands with a bandwidth of 25 Hz. For further details regarding the bandwidth computation, please refer to Appendix E in the paper.
>
> ***Q4 In Figure 2, the authors mention that "Residual connections are used to retain original features and reduce learning difficulty." Could you please indicate where these residual connections are depicted in the figure?***
>
> **A4** We sincerely thank the reviewer for pointing out this issue. We have added Figure 2(a) to help understanding the residual connections in the paper, and represented residual connections in a more understandable way. In 2(a), the residual connections specifically refer to the connections between the yellow block on the leftmost of the diagram and each addition operator. This design implies that the output of each FFI module $Z_{b,t}$ is added to the original features $Z$ and subsequently used as the input for the next FFI module. It is important to emphasize that only the first FFI module takes the original features $Z$ as its input.
>
> ***Q5 In Figure 4, the authors use a line chart to present the results. This may be misleading because the results on the three datasets are independent of each other. A different visualization style, such as a bar chart, might provide a clearer comparison.***
>
> **A5** Thanks for the suggestion. We have represented the result with a bar chart in the paper, see Figure 4 in Section 6.1.
>
> ***Q6 What are the differences between TIGER (small) and TIGER (large), why do they have the same model size?***
>
> **A6** The only difference between TIGER (small) and TIGER (large) is the number of FFI blocks repeated. TIGER (large) has 8 blocks while TIGER (small) has 4 blocks, as described in section 6.2 of the paper. Since all FFI blocks share parameters, TIGER (small) and TIGER (large) have the same model size.
>
> ***Q7 In the ablation study, it looks like the proposed modules appear to involve trade-offs among various performance criteria. Could the authors discuss how to balance this tradeoff and what could be the best to consider?***
>
> **A7** A balance between efficiency and performance should be made based on the specific application scenario. As mentioned in the introduction, speech separation models are often deployed on resource-constrained edge devices in practical applications. Furthermore, as a preprocessing step for speech recognition, low latency is particularly critical for separation models. Therefore, when models demonstrate comparable performance but exhibit significant differences in computational cost (as shown in Table 6 in the paper), we prefer to adopt the MSA structure, which offers higher parallelism and fewer parameters, even if this choice entails a modest trade-off in model performance.
>
> ***References***
>
> [1] Yi Luo, Cong Han, and Nima Mesgarani. “Group Communication With Context Codec for Lightweight Source Separation”. In: IEEE/ACM Transactions on Audio, Speech, and Language Processing 29 (2020), pp. 1752–1761.
>
> [2] Yi Luo and Jianwei Yu. Music source separation with band-split rnn. IEEE/ACM Transactions on Audio, Speech, and Language Processing, 2023.
>
> [3] Philipos C Loizou. Mimicking the human ear. IEEE Signal Processing Magazine, 15(5):101–130, 1998.

---

> ### Author Response · Authors · 2024-11-29
> **Hoping for Further Discussion on Our Response**
>
> Dear Reviewer,
>
> We sincerely thank you for your review and for the valuable feedback you provided. Based on your comments, we have revised our manuscript and submitted detailed responses to your suggestions. We are eager to hear if our updates address your concerns and would welcome the chance for further discussion. Thank you very much in advance.
>
> Authors

---

> > ### Comment · Reviewer_kFvq · 2024-11-30
> >
> > Thank you for the response. The rebuttal partially addressed some of my concerns. I am raising the scoring from 5 to 6.

---

### Official Review · Reviewer_27Ea · 2024-11-07

**Soundness:** 4
**Presentation:** 3
**Contribution:** 4
**Rating:** 8
**Confidence:** 4

**Summary:**

The paper introduces a novel model architecture that has small number of parameters and performs as well as other larger models on various separation tasks.

The model architecture is similar to bandsplit RNN in its encoder/decoder part and in its core part, it has two novel attention-based submodules called MSA and F3A which are applied in the frequency direction and time (frames) direction separately in identical fashion. These modules are repeated a fixed number of times. When repeating, the parameters are tied, so the number of parameters in the model do not increase when the repetition count is increased.

The paper also introduces a new dataset called EchoSet which includes reverberant speech mixtures which improved performance on a "real" dataset. The "real" dataset is not real in the sense of real speech conversations, but it is obtained through playback in a real acoustic environment of individual sources which are mixed later for evaluation.

The performance on real data shown in Figure 4 is much lower compared to the results with artificial mixtures even when training on EchoSet, which shows there is still a lot of room for improvement when applying these models on real data.

**Strengths:**

The paper is well written and the model is described in a detailed manner even though there may be room for improvement in the model description for more clarity.

The method applies to many different sampling rate data due to bandsplit RNN based encoder/decoder. The model is also used to perform cinematic audio separation at 44.1 kHz.

The ablations and various comparisons with state-of-the-art on multiple relevant datasets are all appropriate and impressive.

**Weaknesses:**

Loss function was not mentioned in the main text (or I missed it). Is the loss in (10) in the appendix used for all tasks, or only for cinematic sound separation?

The math in the MSA module description gets a bit hard to follow, so maybe a more detailed Figure that tracks along with mathematical equations would help. How does selective attention (SA) work? It was not described in the paper.

**Questions:**

1. What was the loss function?
2. How does selective attention (SA) work?
3. What is the reasoning behind repeating the same FFI block (with tied parameters)?

---

> ### Author Response · Authors · 2024-11-22
>
> We truly value the time and attention you have dedicated to reviewing our work and providing insightful feedback. Below, we have responded to each of your comments in detail.
>
>
> ***Q1 What was the loss function?***
>
> **A1** Our loss function for training in the speech separation tasks (LRS2-2Mix, Libri2Mix, and EchoSet) was the negative of SI-SNR, which is a commonly used loss function in speech separation [1], as shown in Appendix B in the paper. We have clarified it in the experimental setup (Section 5) in the main text. In the cinematic sound separation task, we used a loss function consistent with the BSRNN to fairly compare the models (as shown in Equation 10).
>
> ***Q2 The math in the MSA module description gets a bit hard to follow, so maybe a more detailed Figure that tracks along with mathematical equations would help.***
>
> **A2** Thanks for the suggestion. We have provided a figure easier to follow in the revised version. Please refer to Figure 3(a). We added some notations of features involved in the fusing stage and decoding stage of the MSA module. Hope this helps understanding.
>
> ***Q3 How does selective attention (SA) work?***
>
> **A3** In the SA module, we fuse global and local signals by the attention mechanism (gate). Specifically, in $f(x,y,z) = \sigma(x) \odot y + z$ (Equation 5 in the paper), $x$ and $z$ are the transformations of global information, while $y$ represents the local information. We first apply the sigmoid function to $x$, generating a value between 0 and 1. Then, the value is used to extract effective features from local information by calculating the element-wise product of $\sigma(x)$ and $y$. Finally, we add the product to $z$, fusing global information and filtered local information.
>
> The mechanism of the SA module indeed needs explanation. We have added it in the paper's method section (Section 3.3.1).
>
> ***Q4 What is the reasoning behind repeating the same FFI block (with tied parameters)?***
>
> **A4** To answer the question of why we share parameters across different FFI blocks, we can draw an analogy to how parameter sharing works in RNNs. In RNNs, parameters are shared across time steps to enable the model to learn patterns in sequences, making it generalizable and efficient. In a similar way, by sharing parameters across FFI blocks, the model can capture repetitive patterns or dependencies within the input data, improving generalization and reducing the overall number of parameters.
>
> ***References***
>
> [1] Le Roux, Jonathan, et al. "SDR–half-baked or well done?." ICASSP 2019-2019 IEEE International Conference on Acoustics, Speech and Signal Processing (ICASSP). IEEE, 2019.

---

> ### Author Response · Authors · 2024-11-29
> **Eager to Hear Your Thoughts on Our Response**
>
> Dear Reviewer,
>
> We sincerely appreciate the effort you invested in reviewing our submission and providing thoughtful comments. We have responded to your suggestions and made revisions to the paper. We would be grateful to hear your thoughts on whether our updates address your concerns and would be happy to discuss further if needed. Thank you again for your time and support.
>
> Authors

---

### Official Review · Reviewer_oAuH · 2024-11-12

**Soundness:** 3
**Presentation:** 3
**Contribution:** 3
**Rating:** 6
**Confidence:** 3

**Summary:**

The paper developed a deep learning model for speech separation; and it not only focuses on improving model performance but model efficiency. The model is lightweight, and is constructed of a new band-split strategy and a new frequency-frame interleaved (FFI) block. Additionally, to reduce the gap between synthetic data and real-world, it introduced a dataset, named EchoSet, with different noise and realistic reverberation. Experimental results demonstrates the effectiveness and efficiency of the proposed model, and the generalization of the dataset to real-world audio.

**Strengths:**

Overall the paper is well-written and easy to follow. The proposed model architecture looks reasonably nice with the introduction of a new band-split strategy and a new FFI block. Experimental results show that the model not only archieves competitive performance on all datasets, but also it is very lightweight in terms of model sizes, MACs. Its efficiency in training (GPU time & memory) and inference (CPU time, GPU time & memory) looks good, too.
The motivation of creating EchoSet dataset is clear and reasonable with a complete analysis of existing dataset. Experimental results on the real-world data with different models, trained on different datasets respectively, demonstrate its generalization ability.

**Weaknesses:**

The FFI block follows a common design of dual-path architecture, it consists of 2 different parts: frequency path and frame path. Each path has two main modules: multi-scale selective attention (MSA) and full-frequency-frame attention (F^3A). While F^3A looks familiar with self-attention mechanism, MSA extracts features through a selective attention mechanism at multiple scales. We may need an ablation study of the MSA architecture with different scales (e.g. 1,2,3,4) to see how it affects model performance.
The evalution on model efficiency is conducted with a fixed audio input length of 1 second. It would be more complete if the evaluation was with different audio input lengths (each audio in EchoSet is 6 seconds, LRS2-2Mix: 2 seconds, Libri2Mix: 3 seconds)

**Questions:**

The model source code is open, the description of EchoSet is reasonable, but it would be better if the configuration/code of data generation is also open.

---

> ### Author Response · Authors · 2024-11-22
>
> We are grateful for your time and the thoughtful feedback you have shared. Below, we have addressed each of your points in detail.
>
> ***Q1 We may need an ablation study of the MSA architecture with different scales (e.g. 1,2,3,4) to see how it affects model performance.***
>
> **A1** The following table shows the ablation study of MSA architecture with scales ranging from 1 to 4. We adopted the small version (4 FFI blocks) of TIGER. Other configurations kept the same as Appendix B and C in the paper. TIGER was trained and tested on EchoSet. Parameters and MACs are evaluated on 1-second input.
>
> | Model         | SDRi (dB) | SI-SDRi (dB) | Params (M) | MACs (G)  |
> |---------------|-------|---------|--------|-------|
> | tiger-scale1  | 10.62 | 10.03   | 0.76   | 10.57 |
> | tiger-scale2  | 13.32 | 12.77   | 0.78   | 9.01  |
> | tiger-scale3  | 13.31 | 12.76   | 0.80   | 8.09  |
> | tiger-scale4  | 13.15 | 12.58   | 0.82   | 7.65  |
>
> According to the table, when the downsampling scale is set to 1, the effectiveness of the selective attention mechanism is limited, leading to a noticeable drop in TIGER's performance. However, with 2, 3, or 4 downsampling layers, TIGER shows comparable performance across configurations, allowing for a suitable model to be selected based on parameters, MACs, and performance trade-offs. This also proves the effectiveness of the selective attention mechanism for audio modeling.
>
> ***Q2 The evalution on model efficiency is conducted with a fixed audio input length of 1 second. It would be more complete if the evaluation was with different audio input lengths (each audio in EchoSet is 6 seconds, LRS2-2Mix: 2 seconds, Libri2Mix: 3 seconds)***
>
> **A2** Thanks for the suggestion. Here, we need to clarify that the input audio length for efficiency evaluation is independent of the audio length. The audio during inference is of variable length when separating audio in the physical world, so there is no one uniform audio length to evaluate model efficiency. Some works evaluated model efficiency also on one-second length [1,2,3,4], others evaluated on four-second segments [5,6]. It is fair to compare the efficiency of these models as long as the same length is used.
> We have supplemented our evaluation results for 6-second audio for reference. The experimental results are shown in the following. As shown in the table, regardless of the duration of audio, TIGER significantly has lower latency and memory usage than models with similar performance.
>
> In the table, the audio length for evaluating efficiency is 1/6 s. 'OOM' stands for out of memory (>24564MB).
>
> | Model       | MACs (G)            | Training GPU Time (ms)   | Training GPU Memory (MB)   | Inference CPU Time (ms)   | Inference GPU Time (ms)  | Inference GPU Memory (MB)   |
> | ----------- | -------------------- | ------------------------ | -------------------------- | ------------------------- | ------------------------- | ---------------------------- |
> | Conv-TasNet | 7.19/43.16           | 92.96/137.32             | 1436.94/8556.42            | 64.21/1325.22             | 13.17/26.53              | 28.78/170.47                |
> | DualPathRNN | 45.01/259.07         | 67.23/296.90             | 1813.55/10242.49           | 723.13/2780.52            | 30.38/157.05             | 298.09/2095.14              |
> | SudoRM-RF   | 4.65/27.91           | 118.46/151.45            | 1353.43/8040.96            | 104.32/571.84             | 20.66/36.15              | 24.42/146.88                |
> | A-FRCNN     | 81.28/487.60         | 230.53/366.78            | 2925.83/16708.81           | 478.58/1840.77            | 82.65/101.19             | 163.82/433.00               |
> | TDANet      | 9.19/54.85           | 263.43/OOM               | 4260.36/OOM                | 434.44/1819.83            | 74.27/120.93             | 136.96/287.84               |
> | BSRNN       | 98.70/587.34         | 258.55/519.65            | 1093.11/10297.47           | 897.27/3821.08            | 78.27/170.56             | 130.24/771.24               |
> | TF-GridNet  | 323.75/2004.02       | 284.17/OOM               | 5432.94/OOM                | 2019.60/11291.58          | 94.30/498.45             | 491.73/4771.31              |
> | TIGER Small | 7.65/56.94           | 160.17/402.09            | 2049.46/11503.89           | 351.15/2226.12            | 42.38/110.29             | 122.23/635.96               |
> | TIGER Large | 15.27/113.73         | 229.23/544.26            | 3989.59/22831.03           | 765.47/3876.61            | 74.51/158.05             | 122.23/640.13               |

---

> ### Author Response · Authors · 2024-11-22
>
> ***Q3 The model source code is open, the description of EchoSet is reasonable, but it would be better if the configuration/code of data generation is also open.***
>
> **A3** Thanks for the suggestion. We provide an anonymous link for hosting the dataset build code (https://anonymous.4open.science/r/TIGER-ICLR2025/echoset-sim.py) and have added this link in the revised version.
>
> ***References***
>
> [1] Li K, Yang R, Hu X. An efficient encoder-decoder architecture with top-down attention for speech separation. arXiv preprint arXiv:2209.15200, 2022.
>
> [2] Yang X, Bao C, Chen X. Coarse-to-fine speech separation method in the time-frequency domain. Speech Communication, 2023, 155: 103003.
>
> [3] Yang X, Ke W, Wei J. Plug-and-play learnable EMD module for time-domain speech separation. Applied Acoustics, 2024, 226: 110210.
>
> [4] Chen C, Yang C H H, Li K, et al. A neural state-space model approach to efficient speech separation. arXiv preprint arXiv:2305.16932, 2023.
>
> [5] Wang Z Q, Cornell S, Choi S, et al. TF-GridNet: Integrating full-and sub-band modeling for speech separation IEEE/ACM Transactions on Audio, Speech, and Language Processing, 2023.
>
> [6] Subakan C, Ravanelli M, Cornell S, et al. Exploring self-attention mechanisms for speech separation. IEEE/ACM Transactions on Audio, Speech, and Language Processing, 2023, 31: 2169-2180.

---

> ### Author Response · Authors · 2024-11-29
> **Looking Forward to Your Feedback on Our Response**
>
> Dear Reviewer,
>
> Thank you so much for taking the time to review our paper and for offering valuable feedback. We have carefully addressed your comments and revised the manuscript accordingly. We are eager to know if we have adequately resolved your concerns and would greatly appreciate the opportunity to discuss further. Thank you again for your time and effort.
>
> Authors

---

### Author Response · Authors · 2024-11-25
**Hope the reviewers will take note of our response**

Dear Reviewers,

After submitting the initial comments, we incorporated your feedback into a revised version of our paper, performed some additional experiments as you requested, and wrote a response to address your main concerns.

We hope to interact with you during the discussion and potentially further improve the quality of our paper.

Thank you very much in advance.

Kind regards,

Authors

---

### Meta-Review · Area_Chair_L8Kh · 2024-12-16

**Metareview:**

The authors presented a deep learning model for speech separation that emphasizes not only improving performance but also enhancing model efficiency. The proposed model is lightweight and incorporates a novel band-split strategy along with a new frequency-frame interleaved (FFI) block. To bridge the gap between synthetic data and real-world scenarios, the authors introduce a dataset called EchoSet, which includes diverse noise types and realistic reverberation. Experimental results demonstrate the model's efficiency.

The key strengths of the proposed solution is the lightweight architecture that allows deployment on resource-constrained edge devices in practical applications. Moreover, the paper is extremely well written and easy to follow. Details of the modules inside the systems are adequately presented. The novel band-split strategy along with a new frequency-frame interleaved (FFI) block.

 A key weakness of the work is that TIGER demonstrates approximately a 6% and 2% performance gap compared to the current SOTA model. Although the authors' point of view is valid, it would have been good to have results on WSJ0-2Mix.

 This work might boost  future research in lightweight speech separation architecture.

**Additional Comments On Reviewer Discussion:**

Six independent reviewers assessed the work. All of them but one were positive about the work. The negative reviewer based their comment mainly on the lack of experiments on WSJ0-2Mix, which cannot be regarded as a major weakness.

 The discussion was very fruitful and key concerns about specific details of the TIGER architectures and additional ablation experiments were resolved by the authors. One of the reviewers increased their score after the discussion period.

---

### Decision · Program_Chairs · 2025-01-22

Accept (Poster)